# Overcoming the Modality Gap in Context-Aided Forecasting

**Vincent Zhihao Zheng** [*1 2]  **Étienne Marcotte** [*1]  **Arjun Ashok** [1 3 4]  **Andrew Robert Williams** [1 3 4]  **Lijun Sun** [2]
**Alexandre Drouin** [1 4]  **Valentina Zantedeschi** [1]

## Abstract

Context-aided forecasting (CAF) holds promise for integrating domain knowledge and forward-looking information, enabling AI systems to surpass traditional statistical methods. However, recent empirical studies reveal a puzzling gap: multimodal models often fail to outperform their unimodal counterparts. We hypothesize that this underperformance stems partly from insufficiently verified context usefulness in existing datasets. To address these limitations, we introduce a semi-synthetic data augmentation method that generates contexts both descriptive of temporal dynamics and verifiably complementary to numerical histories. This approach enables massive-scale dataset creation, resulting in **CAF-7M**, a corpus of 7 million context-augmented time series windows, including a rigorously verified test set. We demonstrate that semi-synthetic pre-training transfers effectively to real-world evaluation, and show clear evidence of context utilization. Our results suggest that dataset quality is a major bottleneck in context-aided forecasting, and that verified context can substantially improve the usefulness of CAF training data.

## 1. Introduction

Time series forecasting is a core component of decision-making across industries, from supply chain optimization to financial planning. While statistical methods like ARIMA (Hyndman & Athanasopoulos, 2018) have been workhorses for decades, *context-aided forecasting* (CAF) has emerged as a promising new paradigm, where external context, often in the form of unstructured textual information (e.g., incident reports, operational notes, expert narratives), is integrated with the numerical histories (Liu et al., 2024a; Williams et al., 2025; Wang et al., 2025a). Such context may include domain knowledge and forward-looking factors (e.g., disruptions, policy changes, campaigns) that are not inferable from historical numerical data alone, yet are essential for accurate predictions.

Despite its enormous potential, recent empirical analyses show that multimodal forecasting models that fuse time series with textual context frequently *fail to outperform unimodal baselines* (Zhang et al., 2025). We hypothesize that this performance gap stems from low context usefulness in both training and test CAF datasets. For a piece of context to be useful, it should be *descriptive* of the underlying dynamics, *aligned* with the temporal window of interest, and *complementary* to patterns already visible in the numerical trajectory. However, obtaining high-quality data at the scale required for training AI models often relies on matching time series with web-scraped text (e.g., news articles) (Liu et al., 2024a; Wang et al., 2024; 2025a), whose predictive utility remains unverified. This quality problem extends even to small-scale test datasets: creating and verifying reliable benchmarks requires human annotators with both domain expertise and time series proficiency to assess whether contexts provide a complementary predictive signal. Without datasets where context is *provably useful*, progress in CAF remains bottlenecked.

To overcome this modality gap, we introduce a generate-then-verify methodology that converts numerical time series datasets into context-aided forecasting datasets with empirically useful context. Our methodology consists of two phases: generation and verification. In the generation phase, we prompt a Large Language Model (LLM) to generate plausible scenarios that explain differences in dynamics between the historical and future portions of a time series window, capturing factors that are not inferable from the historical data alone. In the verification phase, we task a state-of-the-art context-aided forecasting model with predicting the future time series given the history and the generated context. We then keep the forecasting window-context pairs for which the forecast improves when provided with the generated context, compared to without it. This procedure ensures that retained contexts pass a direct forecasting-usefulness

---

[*]Equal contribution  [1]ServiceNow AI Research, Montréal, Canada [2]Department of Civil Engineering, McGill University, Montréal, Canada [3]DIRO, Université de Montréal, Montréal, Canada [4]MILA, Montréal, Canada. Correspondence to: Valentina Zantedeschi <vzantedeschi@gmail.com>.

*Proceedings of the 43rd International Conference on Machine Learning*, Seoul, South Korea. PMLR 306, 2026. Copyright 2026 by the author(s).

test, providing empirical evidence that they are complementary to numerical histories under the chosen verifier.

Critically, our methodology enables **massive-scale dataset creation**. We release CAF-7M[1][2], a semi-synthetic dataset containing 7 million pairs (Sec. 3.2), of which 904 are verified and used for evaluation. We validate this dataset and construction procedure by training DoubleCast, a multimodal architecture that augments the Chronos foundation model (Ansari et al., 2024) with CAF capabilities. Our results are clear: when trained on CAF-7M, DoubleCast (i) effectively leverages context, (ii) achieves performance comparable to state-of-the-art CAF models while being significantly more cost-effective, and (iii) consistently outperforms Chronos, its unimodal counterpart. Further, DoubleCast outperforms both unimodal baselines and state-of-the-art models on the real-world ChatTime benchmark (Wang et al., 2025a), demonstrating that training on semi-synthetic data transfers to real-world context-aided forecasting. Overall, our results demonstrate that the proposed methodology helps bridge the modality gap in time series forecasting, enabling the training of context-aided forecasting models despite the scarcity of strongly relevant contextual information.

**Contributions.**

- A semi-synthetic **methodology for generating verifiably useful contexts** from any time series dataset, enabling massive-scale and high-quality CAF data.
- **CAF-7M**: a corpus of 7 million context-augmented time series windows, spanning 11 domains, including a rigorously verified test set.
- Evidence that our methodology enables **context utilization** and **simulated-to-real transfer**, via a multimodal architecture trained exclusively on CAF-7M.

**Conflict of Interest Disclosure.** Several authors are employed by ServiceNow, whose resources were used to develop and evaluate CAF-7M and DoubleCast. The authors declare no other financial conflicts of interest related to this work.

## 2. Background and Related Work

### 2.1. Problem Setting

We consider the problem of *context-aided forecasting*, where given past observations $z_{1:P} \in \mathbb{R}^P$ of a univariate time series and arbitrary natural language context $\mathbf{x}$, the goal is to estimate a conditional predictive distribution over

---

future values $z_{P+1:P+Q} \in \mathbb{R}^Q$,

$$p(z_{P+1:P+Q} \mid z_{1:P}, \mathbf{x}). \tag{1}$$

A context $\mathbf{x}$ is considered *useful* if conditioning on it improves forecast quality. Formally, let $\mathcal{L}$ be a proper scoring rule and let $z^\star := z^\star_{P+1:P+Q}$ denote a realization drawn from the true future distribution. Context $\mathbf{x}$ is useful if

$$\mathbb{E}_{z^\star}\big[\mathcal{L}(p(\cdot \mid z_{1:P}, \mathbf{x}), z^\star)\big] < \mathbb{E}_{z^\star}\big[\mathcal{L}(p(\cdot \mid z_{1:P}), z^\star)\big], \tag{2}$$

where lower values of $\mathcal{L}$ indicate better forecasts.

### 2.2. Multimodal Time Series Datasets

Several datasets for context-aided forecasting have been proposed, but ensuring context relevance remains challenging due to low signal-to-noise ratios in naturally available sources. One approach leverages domains where timestamped articles are naturally aligned with time series, as in FinMultiTime (Xu et al., 2025), MoTime (Zhou et al., 2025), and MTBench (Chen et al., 2025); to reduce input length, some use LLMs to summarize lengthy reports (Liu et al., 2024a). A second approach uses LLMs to generate descriptions from numerical patterns in the historical window, as in TGTSF (Xu et al., 2024) and TSFragment-600K (Ge et al., 2025). Finally, template-based methods convert metadata or summary statistics into text (Wang et al., 2025a; Xu et al., 2024; Williams et al., 2025). Table 1 compares these datasets along key dimensions.

In line with previous work, our approach uses LLMs to generate scenarios aligned with time series windows, leveraging their broad background knowledge to produce diverse and plausible textual context. Unlike prior methods that condition only on historical values or restrict the impact of future values to rigid templates, we allow the LLM to be flexible in how it extracts information from both history and future values. This enables the LLM to generate context that is both genuinely informative for prediction while still being plausible. Aurora is the closest concurrent work on large-scale multimodal pretraining for probabilistic time series forecasting (Wu et al., 2026). Our approach is complementary: beyond generating context for training, our generate-then-verify procedure retains evaluation instances only when the generated context improves forecasts over a matched no-context baseline.

### 2.3. Models for Context-Aided Forecasting

Recent work in context-aided forecasting leverages language models to combine contextual and numerical data, following two paradigms (Zhang et al., 2025): (i) *alignment-based* methods, i.e. multimodal architectures that fuse time-series representations with text embeddings (Jin et al., 2024; Liu et al., 2024b; Wang et al., 2025a), or (ii) *prompt-based* methods, which are LLM-based forecasters that

*Table 1.* Comparison of dataset size in terms of domains and textual contexts. We distinguish between independent texts and template-generated texts. Further, we emphasize datasets that are LLM-generated and provide only evaluation data. CAF-7M contains nearly twice as many texts as any prior dataset while covering more diverse domains and providing both training and evaluation data.

| Dataset | # Domains | # Ind. texts | # Templates | # Templated texts | LLM Generated | Test only |
|---|---|---|---|---|---|---|
| **Time-MMD** (Liu et al., 2024a) | 9 | 26,880 | – | – | ✓ | |
| **FinMultiTime** (Xu et al., 2025) | 1 | 3,742,445* | – | – | | |
| **MoTime** (Zhou et al., 2025) | 4 | 910,908 | – | – | | |
| **CGTSF** (Wang et al., 2025a) | 3 | – | 3 | 33,693 | | |
| **MTBench** (Chen et al., 2025) | 2 | 4,026* | – | – | | |
| **TGTSF** (Xu et al., 2024) | 2[†] | 87,160[‡] | 3 | 162* | ✓ | |
| **TSFragment-600K** (Ge et al., 2025) | 5 | 700,800 | – | – | ✓ | |
| **Context is Key (CiK)** (Williams et al., 2025) | 8 | – | 71 | 355 | | ✓ |
| **CAF-7M (Ours)** | 11 | 7,433,239 | – | – | ✓ | |

∗: These datasets contained many duplicated entries. These numbers only count non-duplicated texts.
†: Two of the datasets mentioned in the paper were not publicly available at time of writing.
‡: This number reflects the amount of LLM generated snippets which are then combined according to a template.

prompt large language models with numerical and textual inputs (Xue & Salim, 2024; Gruver et al., 2023; Requeima et al., 2024; Williams et al., 2025). A major drawback of prompting-based methods is their computational cost. For example, the Direct Prompt method from Williams et al. (2025) that we evaluate in Sec. 4.1 requires many tokens to generate a single floating-point number. In Section 3.3, we introduce a distinct alignment-based method that leverages a specialized dual-attention decoder. Appendix D contains an extended review of both alignment-based methods and prompt-based methods.

## 3. Proposed Method

### 3.1. Data Augmenting Time Series Datasets

A trivially useful context $\mathbf{x}$ would directly describe $z_{P+1:P+Q}$ in textual form, allowing exact recovery of the future values. However, such context is unrealistic, and a model trained this way would not generalize to real-world data. Instead, we aim for context that is both plausible and informative. As shown in Fig. 1, our pipeline selects a time series dataset as a starting point, generates plausible context, and filters for quality.

**Generating Plausible Contexts**   Large language models have been shown to be accurate context-aided forecasters, even without specific training for the task (Williams et al., 2025). This raises a natural question: can we also use them to generate context for a given observation $z_{1:P+Q}$? We do so by prompting the model with both historical and future values, along with a short description of the data domain drawn from the original publications. This future conditioning is intentional: it is used only as an offline data-construction mechanism to simulate forecast-time side information and test whether models exploit it, not as an online forecasting protocol. We then instruct the model to produce a scenario that would help predict the future values, focusing in particular on changes in dynamics from the historical series. However, many LLMs produce data

of limited diversity, even at higher temperatures, resulting in redundant contexts. To address this, we take inspiration from Merrill et al. (2024) by prompting the model to avoid repeating scenarios from previously generated contexts. See App. A.1 for the prompt.

**Ensuring Informative Contexts**   While LLM-generated contexts can appear informative at first glance, they may fail to introduce new information, contain incorrect information, or be too difficult for state-of-the-art context-aided models to exploit. To ensure quality, we first perform a basic check by prompting an LLM to verify whether the context explicitly mentions the variable represented in the time series and whether the scenario is plausible for that variable. See App. A.1 for the prompt used; as a complementary surface-quality sanity check, 11 annotators evaluated 30 generated contexts, with 80% clear-quantity and 82% plausible-scenario majority votes (Tab. 6).

We then filter contexts by checking their compliance with Eq. (2), i.e., does including the context improve the forecast? We use a Direct Prompt forecaster (Williams et al., 2025) with a strong LLM (e.g., GPT-5.2) as the distribution estimator because it can be applied zero-shot with and without context under the same forecasting protocol. This makes it a practical verifier for context usefulness, whereas multimodal TSFMs typically require task-specific training and may introduce their own context-style or architecture-specific biases. We use the Continuous Ranked Probability Score (CRPS; Gneiting & Raftery (2007)) as the proper scoring rule. Samples for which the CRPS with context exceeds the CRPS without context are discarded, considering that the context may be insufficiently informative. The rest of the samples for which the forecast of the model improves with the generated context are kept. This provides a set of contexts that are complementary to numerical histories, accounting for dynamic shifts and providing external predictive cues.

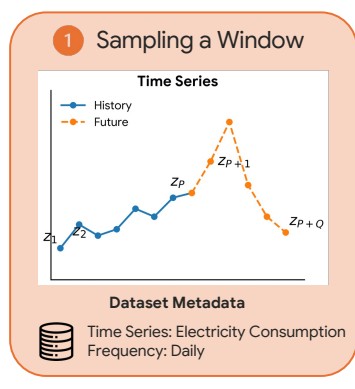 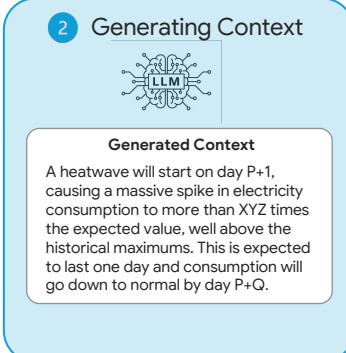 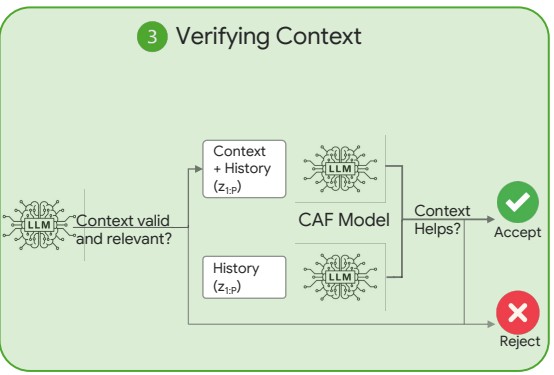

*Figure 1.* The data-augmentation pipeline: (1) From each source dataset, we sample forecasting windows consisting of a numerical history and prediction horizon, along with dataset metadata. (2) We generate scenario-style textual context conditioned on the window. (3) We verify context relevance by checking whether a strong CAF method achieves better predictions with context than without (e.g., lower CRPS); windows are *accepted* if context improves predictions and *rejected* otherwise.

## 3.2. CAF-7M: A Context-Aided Forecasting Dataset

We now introduce **CAF-7M** (**C**ontext-**A**ided **F**orecasting), a large-scale context-aided forecasting corpus built using the methodology in Sec. 3.1. CAF-7M contains 7,433,239 context-augmented time series windows for training and 904 for testing. The numerical data for these windows are sampled from 65 real-world datasets spanning diverse domains, with 40 used for training and 25 reserved for testing (see Tab. 4). We follow the dataset selection and train/test split from Chronos (Ansari et al., 2024) to ensure domain diversity and comparability. The number of windows from each dataset is detailed in App. A.2. Prediction horizons are set according to data frequency (e.g., 64 for hourly and daily; see App. A.1), and history lengths are sampled uniformly between the prediction horizon and 512.

The raw numerical time series windows are then augmented with synthetic contexts generated according to Sec. 3.1. We use Llama-3.3-70B-Instruct (Grattafiori et al., 2024) to generate contexts and GPT-5.2 (OpenAI, 2025) as the backbone for the Direct Prompt forecaster used in verification. For cost reasons, the forecaster-based filtering is only applied to the testing set, as probabilistic forecasting at this scale is prohibitively expensive. This yields unfiltered training data but rigorously validated testing data, which is arguably more critical for evaluation.

We partition the test samples into two subsets, according to their difficulty, and encourage practitioners (i) to explicitly mention which subset is used in their evaluation and (ii) not to randomly sample from the full set. Indeed, one risk of sampling uniformly is that the resulting test windows may be too easy to forecast without context. Such a dataset would poorly measure a model's ability to exploit context, as only marginal improvements would remain possible. Conversely, windows that are extremely challenging for unimodal models would be unrepresentative of real-world use cases. To balance both extremes, we split the testing set according to

Chronos's unimodal forecasting performance, using a Mean Absolute Scaled Error (MASE) threshold of 1.5. Windows with MASE above 1.5 form the HARD split; those below form the EASY split. We sample an equal number from each before context augmentation. After filtering, we obtain 490 HARD and 414 EASY windows, for a total of 904 testing windows (see App. A).

## 3.3. Training a Context-Aided Forecasting Model

To validate that CAF-7M can serve as an effective context-aided forecasting training dataset, we train DoubleCast as an alignment-based forecasting model for measuring whether models can exploit instance-aligned textual context (Zhang et al., 2025). Unlike broader multimodal forecasting systems such as Aurora (Wu et al., 2026), DoubleCast is intended primarily as a controlled vehicle for evaluating CAF-7M rather than as an exhaustive architecture search.

**Aligning Context and Time Series** There are many ways to combine the two modalities architecturally, such as concatenation, addition, or attention, and each with different capacity and computational trade-offs. A simple alignment-based fusion is as follows. A time-series encoder $\phi_z : \mathbb{R}^P \to \mathbb{R}^h$ maps the history $z_{1:P}$ to an $h$-vector $\mathbf{e}_{ts}$. A text encoder $\phi_x : \mathcal{X} \to \mathbb{R}^{h_x}$ embeds the unstructured context $\mathbf{x}$ into $\mathbf{e}_{ctx}$. We then fuse by concatenation via $\psi : \mathbb{R}^h \times \mathbb{R}^{h_x} \to \mathbb{R}^{h+h_x}$. Finally, a forecaster $g : \mathbb{R}^{h+h_x} \to \mathcal{P}(\mathbb{R}^Q)$ outputs the joint distribution over future values:

$$p\big(z_{P+1:P+Q} \mid z_{1:P}, \mathbf{x}\big) \approx g\big(\psi\big(\phi_z(z_{1:P}), \phi_x(\mathbf{x})\big)\big). \quad (3)$$

By pretraining on large corpora of contextual scenarios generated by LLMs, this approach learns to align textual context with temporal dynamics, producing forecasts that adapt dynamically to the conditions encoded in $\mathbf{x}$.

**DoubleCast** We introduce an alignment-based method which we call *DoubleCast*. DoubleCast synergizes the

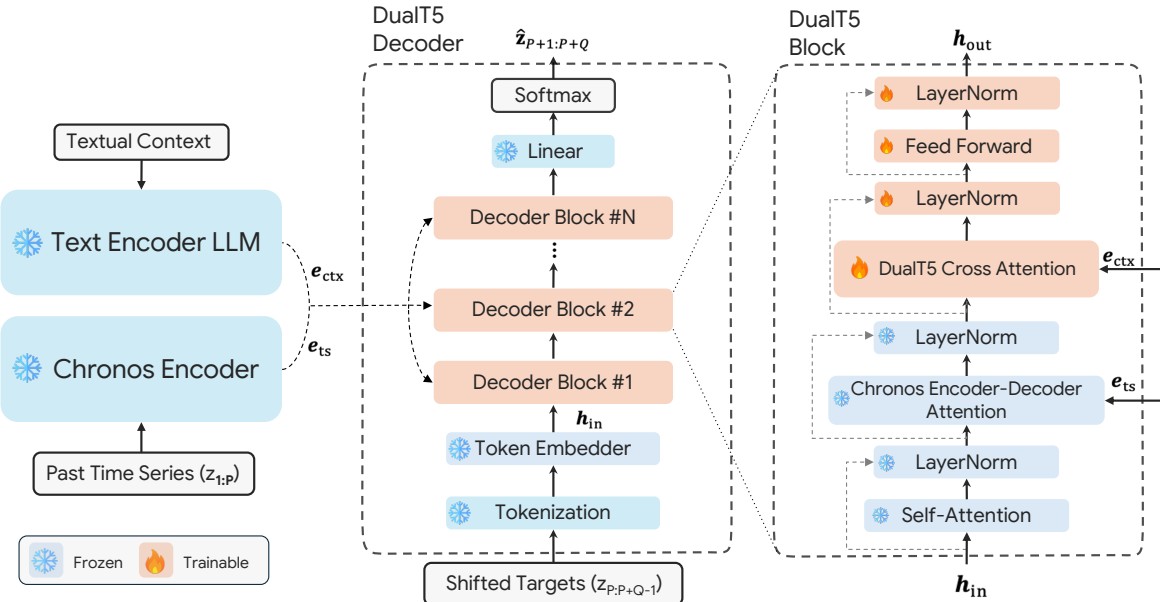

*Figure 2.* **Architecture of DoubleCast**. Each DualT5 decoder block consists of, in sequence: masked self-attention; Chronos encoder–decoder cross-attention; DualT5 cross-attention; and a FFN layer. Each sublayer is wrapped by a residual connection and layer normalization (LN). The same $e_{ts}$ and $e_{ctx}$ are provided to every decoder block.

Chronos time series foundation model (Ansari et al., 2024) with an independent pre-trained text encoder to condition forecasts on unstructured text. The architecture consists of three primary components:

1. A **Time-Series Encoder**, which processes the historical time-series into a sequence of embeddings, $e_{ts}$. This component is inherited directly from Chronos or another Time Series Foundation Model.
2. A **Text Encoder**: a pre-trained LLM (e.g., Qwen (Yang et al., 2025)) that encodes the context into a sequence of high-dimensional hidden states $e_{ctx}$, which may be extracted from any chosen hidden layer.
3. A **DualT5 Decoder**, a decoder architecture which autoregressively generates the forecast by attending to both the time-series and text embeddings.

DoubleCast distinguishes itself by its specialized decoder: each Chronos decoder block is augmented with an additional *DualT5 cross attention layer* placed above the original Chronos encoder–decoder attention. This secondary cross-attention mechanism enables the decoder to integrate signals from the text encoder to condition on natural language context. Fig. 2 illustrates the whole architecture.

The central claims of our paper focus on the role of context in context-aided forecasting, not on the architecture. To keep the experimental focus on the data, we adopt a single, principled architectural configuration. Specifically, we use `Qwen3-14B` as the text encoder to balance quality and efficiency, extract embeddings from the second-to-last layer, and apply text-conditioning in every decoder block.

Appendix B gives more detailed information on DoubleCast architecture and App. C.1 explains how it was trained.

## 4. Experiments

We envision CAF-7M as primarily a training dataset. Its usefulness is thus directly linked to whether pretraining a context-aided forecasting model using CAF-7M is a good starting point or not. To validate its utility, we ideally would train models on CAF-7M and evaluate them on diverse real-world benchmarks. However, when models underperform on existing datasets, we cannot distinguish whether the issue lies in their inability to use context or in the context lacking predictive value. To address this ambiguity, we first evaluate on CAF-7M's verified test sets, where we know the contexts are useful, before assessing transfer to the real-world ChatTime benchmark (Wang et al., 2025a).

We organize our analysis around three questions:

- **RQ1: Does CAF-7M's testing set contain informative contexts, complementary to time series?**
- **RQ2: Does training on CAF-7M enable generalization to its testing set?**
- **RQ3: Does training on CAF-7M enable generalization to real-world context-aided forecasting datasets?**

### 4.1. Context Complementarity in the CAF-7M Test Set

We first validate whether contexts in CAF-7M's test set contain complementary predictive signal, by assessing whether they provide measurable forecasting improvements. We apply the Direct Prompt forecasting technique (Williams et al.,

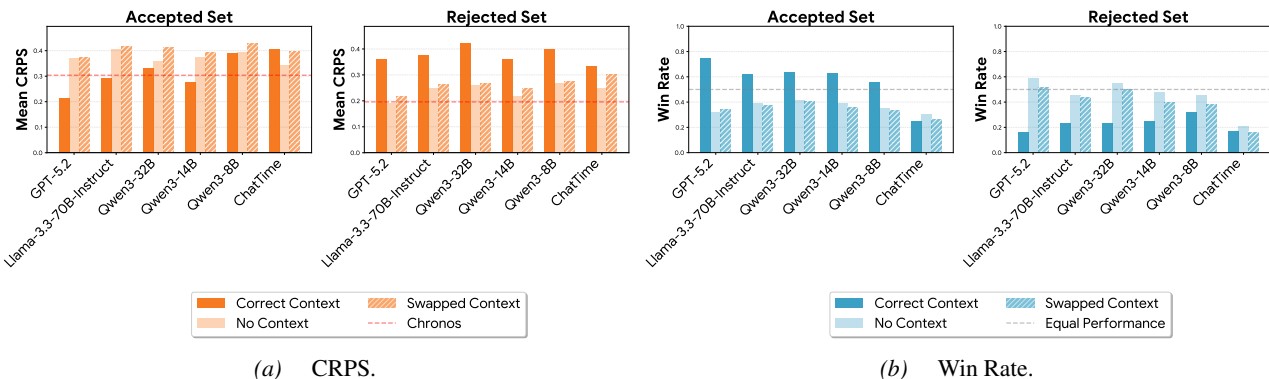

*(a)* CRPS.                    *(b)* Win Rate.

*Figure 3.* CRPS (↓) and Win Rate (↑) for Direct Prompt and ChatTime on the **HARD** split (Tab. 2), filtered into Accepted and Rejected Sets by GPT-5.2 (Sec. 3.1). We assess each method with (i) correct context, (ii) no context, and (iii) swapped context from another instance. Filtering is crucial: larger models benefit (worsen) more from context on the Accepted (Rejected) Set. The usefulness of the context also generalizes to LLMs beyond GPT-5.2, with larger models benefiting more (in line with Zhang et al. (2025)). Therefore, we use Qwen3-14B for DoubleCast (Sec. 3.3) since it balances cost and performance.

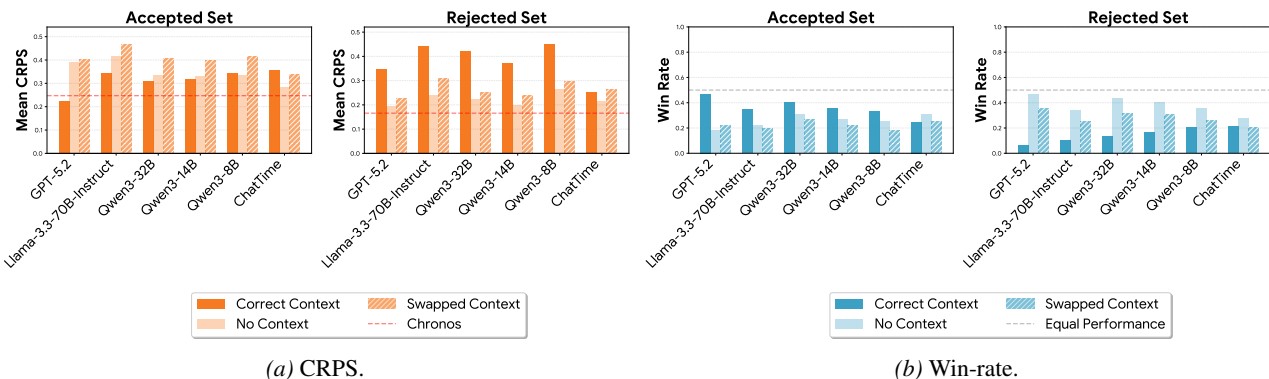

*(a)* CRPS.                    *(b)* Win-rate.

*Figure 4.* CRPS (↓) and Win Rate (↑) for Direct Prompt and ChatTime on the **EASY** split, using the same accepted/rejected and correct/no-context/swapped-context comparison as in Fig. 3.

2025) with a variety of generic LLMs, zero-shot, and with or without context (see App. E.4 for the full prompt). If the contexts do indeed contain complementary information to the historical time series data, then the forecasts generated by Direct Prompt will be much more accurate when the context is provided than when it is not.

Since Direct Prompt produces a probabilistic forecast, we use the CRPS (Gneiting & Raftery, 2007) as our primary metric, which is a proper scoring rule for probabilistic forecasts. To avoid large values dominating the aggregated results, we normalize the CRPS using the mean absolute values of the forecast ground truths. As a complementary metric that is robust to score outliers, we also compute the *Win Rate*: the ratio of windows where Direct Prompt has a lower CRPS than Chronos.

Figure 3 shows the impact on the forecasting accuracy on the HARD split's accepted and rejected sets (Sec. 3.1) when context is withheld (No Context) or mismatched between samples (Swapped Context). Contexts in the accepted set

lead to a significant improvement both in terms of CRPS and of Win Rate for the Direct Prompt method with any LLM backbone. Notably, mismatched contexts (Swapped Context) do not improve performance over providing no context, showing that Direct Prompt does not benefit from any random context. Together, these observations confirm that performance gains on the accepted set are driven by *complementary information in the contexts*. Finally, as samples are filtered with GPT-5.2, the superior performance of this backbone could be more indicative of bias to the verifier than actual superior forecasting.

The importance of the filtering procedure described in Sec. 3.1 is revealed when we compare the results for the Accepted Set and the Rejected Set in Fig. 3: in the Rejected Set, the accuracy of the forecasts is worse when the context is given than when it is missing. This highlights how important it is to filter synthetic contexts since they could lead to the wrong conclusion that a forecasting method is unable to leverage context, when in reality the contexts are simply of too low quality to be useful. Note that even though sample

selection is done using GPT-5.2, its results are highly correlated with those for the other models, both for the accepted and the rejected sets. This suggests that accepted samples are not overly tailored to a single LLM, but rather broadly contain useful predictive information that generalizes across forecasters.

The EASY split shows weaker but consistent trends, as expected when numerical histories leave less room for context to help (Fig. 4); appendix verifier-choice analyses further show that DP acceptance depends on the forecasting model, with GPT-5.2 agreeing most with the best-case verifier (App. F.6). Together, these results clarify that verification is not merely a plausibility filter for generated text, but a forecasting-based selection mechanism for measuring context usefulness under controlled ablations.

### 4.2. Benchmarking on CAF-7M Test Set

To assess whether training a context-aided forecasting model on the training set of CAF-7M leads to a model which generalizes to its verified test set, we train both TimeLLM (Jin et al., 2024) and DoubleCast on it, and compare their accuracy against multiple forecasting models run zero-shot: Chronos, versions of TimeLLM pre-trained on other datasets, ChatTime, and Direct Prompt using either GPT-5.2 or Qwen3-14B (see App. E for more details).

Tab. 2 reports the different aggregated CRPS and Win Rates for these various baselines and the models trained on CAF-7M. We observe that DoubleCast has a very strong performance, being second only to Direct Prompt with GPT-5.2, which has the advantage of having been used to filter the test samples. This indicates that while filtering is important for constructing a reliable test set, training on CAF-7M leads to performant context-aided forecasting models, even though the training samples were not forecaster-filtered (Sec. 3.1). This train-test quality mismatch reflects a practical quantity-quality trade-off: DP verification is too expensive for all 7M training windows, and although filtered subsets are more sample-efficient at matched budgets, the gap narrows as the unfiltered budget increases (Tab. 5).

To check whether models trained on CAF-7M make use of the context, Tab. 2 also reports by how much the CRPS and Win Rate increase or decrease when the context is either omitted (no context) or randomly exchanged between windows (swapped). We see that DoubleCast actively uses the context since both metrics are markedly worse when context is missing or swapped, with the CRPS increasing from 11.1% to 26.5% and the Win Rate decreasing from 21.5% to 41.9%. This conclusion does not apply to TimeLLM trained on CAF-7M, since some of its results improve when the context is removed or swapped. This suggests that TimeLLM was unable to learn to properly use the context in CAF-7M, even though its results are better when trained on CAF-7M

than on its original training data. While dataset scale is critical, these results highlight that model architecture plays an equally decisive role; further analysis would be required to assess if this failure to exploit high-quality context extends to other datasets.

### 4.3. Generalization to Real-World Benchmarks

We finally assess whether a model trained on CAF-7M can generalize to other real-world datasets that do not share the same domains or structure. To that end, we use a test split from the CGTSF dataset (Wang et al., 2025a) as an out-of-domain benchmark for DoubleCast. We compare 4 models on the 3 CGTSF datasets: Chronos, as the unimodal version of DoubleCast; DoubleCast solely trained on CAF-7M, to determine its zero-shot capabilities; DoubleCast pre-trained on CAF-7M and finetuned on CGTSF, to see if it can serve as a good starting point for finetuning; and finally the official release of ChatTime, which was trained on CGTSF.

In Tab. 3, we observe that the finetuned version of DoubleCast has better performance than both Chronos and ChatTime on all but one of the dataset-prediction length pairs. This consistent advantage provides strong evidence that CAF-7M serves as an effective pretraining dataset with demonstrated simulated-to-real transfer.

The results for DoubleCast solely trained on CAF-7M are nuanced: while it is always more accurate than ChatTime, it is only better than Chronos on LEU and PTF, falling short on MSPG. Since the gap with the finetuned results is smaller than with the Chronos results on LEU and PTF, we conclude that DoubleCast trained on CAF-7M is able to zero-shot leverage the contextual data of these splits, but that data has limited complementarity with the historical time series. As for MSPG, the small gap between Chronos and finetuned DoubleCast indicates that the contexts contain limited auxiliary information for the forecast.

In contrast to LEU and PTF, MSPG requires 96-step predictions, which exceeds CAF-7M's 64-step maximum. When Chronos and DoubleCast forecast on data with a prediction length greater than 64, the forecasting is done autoregressively on chunks of 64 time steps. This means that the context is given to DoubleCast twice, while it was not trained to apply the context on the second call (time steps 65 to 96). This distribution mismatch explains the poor zero-shot MSPG performance, making the finetuned model's success particularly striking: it learns to extend context application beyond its pretraining horizon.

We further evaluate DoubleCast on Time-MMD (Liu et al., 2024a); detailed results are reported in App. F.3. Fine-tuned DoubleCast improves over zero-shot DoubleCast on all nine domains by MAE and outperforms Chronos on seven of nine domains. Additional appendix results include Chronos-

*Table 2.* Benchmarking on test set of CAF-7M, using both CRPS and Win Rate. The results are for both the splits of the testing set (HARD and EASY) and the combination of both (ALL). The methods are split between zero-shot methods (top) and those trained on CAF-7M (bottom). "No context" refers to evaluation on the same series without the context, while "swapped" refers to evaluation of the same series with context from another instance. The best results are bolded and the second best results are underlined.

| | ALL | | HARD | | EASY | |
|---|---|---|---|---|---|---|
| MODEL | CRPS | WIN | CRPS | WIN | CRPS | WIN |
| Chronos | $0.278 \pm 0.002$ | N/A | $0.304 \pm 0.002$ | N/A | $0.247 \pm 0.004$ | N/A |
| TimeLLM (ETTm1, 192) | $0.635 \pm 0.001$ | 0.08 | $0.549 \pm 0.001$ | 0.12 | $0.738 \pm 0.001$ | 0.03 |
| TimeLLM (ETTm1, 336) | $3.309 \pm 0.003$ | 0.00 | $3.212 \pm 0.004$ | 0.00 | $3.410 \pm 0.005$ | 0.00 |
| TimeLLM (ETTm2, 336) | $0.803 \pm 0.001$ | 0.05 | $0.703 \pm 0.001$ | 0.07 | $0.921 \pm 0.001$ | 0.02 |
| TimeLLM (ETTm2, 96) | $1.406 \pm 0.002$ | 0.01 | $1.284 \pm 0.002$ | 0.02 | $1.546 \pm 0.003$ | 0.00 |
| ChatTime | $0.385 \pm 0.001$ | 0.25 | $0.407 \pm 0.001$ | 0.25 | $0.358 \pm 0.002$ | 0.25 |
| DP GPT-5.2 | $\mathbf{0.217 \pm 0.001}$ | 0.62 | $\mathbf{0.212 \pm 0.001}$ | 0.75 | $0.223 \pm 0.001$ | 0.47 |
| DP Qwen3-14B | $0.296 \pm 0.001$ | 0.51 | $0.277 \pm 0.002$ | 0.63 | $0.319 \pm 0.002$ | 0.36 |
| TimeLLM (CAF) | $0.501 \pm 0.001$ | 0.17 | $0.428 \pm 0.001$ | 0.22 | $0.587 \pm 0.001$ | 0.10 |
| TimeLLM (CAF) (no context) | -8.0% | -26.7% | -1.5% | -34.5% | -14.0% | -11.9% |
| TimeLLM (CAF) (swapped) | -1.1% | +8.7% | +2.0% | +7.3% | -3.6% | +7.1% |
| DoubleCast | $0.231 \pm 0.001$ | **0.71** | $0.251 \pm 0.001$ | **0.79** | $\mathbf{0.204 \pm 0.002}$ | **0.64** |
| DoubleCast (no context) | +18.3% | -34.7% | +22.0% | -41.9% | +11.1% | -21.5% |
| DoubleCast (swapped) | +26.5% | -37.6% | +25.2% | -40.8% | +25.4% | -37.7% |

2, FlowState, TimeMixer++, Multi-Patch Prediction, and Aurora. Aurora also degrades under shuffled context, but less strongly than DoubleCast on the same test set: CRPS increases by 9.2% for Aurora versus 26.5% for DoubleCast on the ALL split, while Fig. 3 shows the same aligned-versus-missing/mismatched context pattern for DP GPT-5.2. Time-MMD remains strongly influenced by the unimodal forecasting backbone, making it useful as an external benchmark but less diagnostic of localized context usage.

**Performance on Standard Numerical Forecasting Tasks**
A crucial concern for CAF-7M is whether training a model for contextual awareness degrades the model's numerical forecasting performance (i.e. in forecasting tasks with no context). To assess this, we evaluate DoubleCast on GIFT-Eval, a purely numerical time series forecasting benchmark (Aksu et al., 2024) that is designed to measure general-purpose forecasting performance. Fig. 5 shows that DoubleCast retains Chronos' forecasting abilities: while performance degrades slightly, DoubleCast and Chronos perform almost identically when compared against AutoArima and Migas-1.0, the current leading statistical and pre-trained models on GIFT-Eval.

## 5. Limitations

**Verifier bias and diversity limits at scale.** Our methodology pairs TS windows with LLM-generated textual scenarios, and constructs a curated evaluation split using an LLM verifier: a window is *accepted* if adding the generated context reduces the verifier's CRPS, and *rejected* otherwise. This procedure yields a benchmark with measurable context complementarity, but can bias the accepted subset toward context patterns that the specific verifier can exploit, while

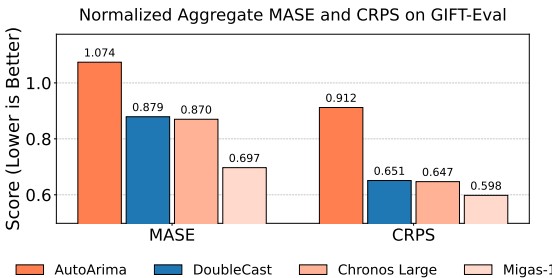

*Figure 5.* Normalized MASE (left, ↓) and CRPS (right, ↓) on GIFT-Eval (Aksu et al., 2024). Despite extending Chronos for context-aided forecasting, DoubleCast retains Chronos' forecasting capabilities: DoubleCast performs nearly identically to Chronos on no-context time series forecasting, as compared to both AutoArima and the incumbent state-of-the-art pre-trained model (Migas-1.0).

potentially under-representing windows where context is helpful in ways not captured by the verifier. In App. F.6, GPT-5.2 shows the highest agreement with a best-case verifier baseline among the models tested, supporting its use as the strongest empirical verifier available to us, while leaving verifier-specific selection effects as a limitation. Moreover, verifier-based filtering is expensive and therefore not applied to the 7M-window training corpus, leaving training contexts potentially noisy or weakly aligned. Because these contexts are generated from future trajectories rather than observed real-world reports, CAF-7M should be interpreted as an offline benchmark for context use rather than a real-time data-availability setting; the main concern is ecological validity, since generated scenarios may not match the form, availability, or reliability of deployment context. Finally, generating textual scenarios at massive scale can exhibit limited diversity (e.g., repetitive phrasing or overuse of common event templates) despite explicit diversity prompt-

*Table 3.* CRPS (↓) on real-world CGTSF datasets, which have been used to train ChatTime (Wang et al., 2025a), for various history and prediction lengths. We show results for both DoubleCast trained solely on CAF-7M and DoubleCast pretrained on CAF-7M before being finetuned on the training split of CGTSF. The best and second best CRPS values for each dataset are bolded and underlined, respectively. DoubleCast without finetuning outperforms Chronos and ChatTime on LEU and PTF, but struggles with MSPG's prediction length of 96, which exceeds CAF-7M's maximum of 64 (indicated by the *). However, DoubleCast finetuned on CGTSF performs best in most settings, demonstrating that training on CAF-7M leads to context-processing capabilities that transfer to real-world datasets.

| DATASET | HIST | PRED | CHRONOS | DOUBLECAST | DOUBLECAST (FT. ON CGTSF) | CHATTIME |
|---|---|---|---|---|---|---|
| PTF | 48 | 24 | $0.1598 \pm 0.0009$ | $0.1534 \pm 0.0007$ | $\mathbf{0.1517 \pm 0.0008}$ | $0.2073 \pm 0.0004$ |
| | 72 | 24 | $0.1401 \pm 0.0008$ | $\underline{0.1355 \pm 0.0007}$ | $\mathbf{0.1336 \pm 0.0007}$ | $0.1816 \pm 0.0005$ |
| | 96 | 24 | $0.1343 \pm 0.0007$ | $\underline{0.1294 \pm 0.0007}$ | $\mathbf{0.1279 \pm 0.0007}$ | $0.1693 \pm 0.0005$ |
| | 120 | 24 | $0.1297 \pm 0.0007$ | $\underline{0.1236 \pm 0.0007}$ | $\mathbf{0.1234 \pm 0.0007}$ | $0.1478 \pm 0.0004$ |
| LEU | 96 | 48 | $0.4646 \pm 0.0012$ | $\underline{0.4585 \pm 0.0011}$ | $\mathbf{0.4566 \pm 0.0012}$ | $0.4892 \pm 0.0006$ |
| | 144 | 48 | $0.4562 \pm 0.0017$ | $\underline{0.4497 \pm 0.0012}$ | $\mathbf{0.4492 \pm 0.0011}$ | $0.4826 \pm 0.0006$ |
| | 192 | 48 | $0.4504 \pm 0.0014$ | $\underline{0.4461 \pm 0.0009}$ | $\mathbf{0.4456 \pm 0.0011}$ | $0.4740 \pm 0.0007$ |
| | 240 | 48 | $0.4516 \pm 0.0018$ | $\mathbf{0.4434 \pm 0.0009}$ | $\underline{0.4437 \pm 0.0014}$ | $0.4640 \pm 0.0006$ |
| MSPG | 192 | 96* | $\mathbf{0.5568 \pm 0.0026}$ | $0.7683 \pm 0.0024$ | $\underline{0.6173 \pm 0.0025}$ | $0.9746 \pm 0.0010$ |
| | 288 | 96* | $\underline{0.4693 \pm 0.0023}$ | $0.6054 \pm 0.0025$ | $\mathbf{0.4649 \pm 0.0023}$ | $0.9679 \pm 0.0010$ |
| | 384 | 96* | $\underline{0.4359 \pm 0.0023}$ | $0.5675 \pm 0.0025$ | $\mathbf{0.4169 \pm 0.0021}$ | $0.9661 \pm 0.0011$ |
| | 480 | 96* | $\underline{0.4191 \pm 0.0023}$ | $0.5160 \pm 0.0024$ | $\mathbf{0.3891 \pm 0.0022}$ | $0.9655 \pm 0.0011$ |

ing, which reduces the effective information content of the contextual modality and may encourage shallow keyword-to-pattern correlations. While our generalization results (Sec. 4.3) show that CAF-7M displays simulated-to-real transfer, these issues could be addressed by running efficient verifiers, such as strong alignment-based forecasters, as well as improved generation strategies (e.g., mixture-of-prompts, retrieval augmentation, or explicit diversity constraints).

**Ecological Validity.** We construct contexts that are forward-looking: they are generated based on the future window. They are therefore cleaner, more on-topic, and more consistently structured than the context a deployed forecaster would encounter. Our results should thus be read as a probe of whether models can exploit free-form, forward-looking context at all, rather than as an estimate of the gains achievable under realistic deployment conditions. Closing this gap will require evaluating context-aided forecasting with naturally sourced contexts, with varying degrees of relevance, reliability, conflict and signal-to-noise ratio.

**DoubleCast's failure modes.** DoubleCast conditions a Chronos-style decoder on text via additional cross-attention layers with learned projections. This is lightweight and effective, but it does not explicitly enforce fine-grained, time-local grounding between specific phrases and forecast steps. As a result, the model can fail in cases where the context specifies *precise temporal localization* (e.g., "a disruption occurs during the first two days of the horizon" or "a rebound begins around week 6") and accurate forecasting requires mapping that description to an exact region of the prediction window. Without an explicit mechanism to bind textual time references to forecast indices (e.g., event-time tags, time-aligned token routing, or a structured event representation), cross-attention may diffuse the signal across the

horizon, leading to smeared effects (e.g., anticipating the change too early, delaying it, or spreading it over too many steps). This limitation is amplified when contexts contain multiple time-scoped events or when the horizon is long relative to the stated time frame, making it challenging to "locate" the relevant context region reliably.

## 6. Conclusion

We demonstrate that using LLMs to augment real-world time series with forward-looking synthetic context provides a viable solution to the lack of high-quality context-aided forecasting datasets. We do so by introducing **CAF-7M**, a 7-million-window corpus of scenario-style contexts aligned to forecasting windows. We also design an extensive verification pipeline to ensure that the 904 instances in the test split of CAF-7M contain context relevant for context-aided forecasting. Our results show that our data generation procedure combined with the verification procedure leads to contexts that are highly complementary to the time series data. To complete this empirical analysis we introduce DoubleCast, a new multi-modal model architecture, and show that when trained on the CAF-7M training split, it not only performs competitively on the CAF-7M test split, but also generalizes to a real-world benchmark. This work highlights two practical conclusions: (i) meaningful progress in context-aided forecasting benefits from benchmarks that explicitly test context usefulness under controls, and (ii) architectures that enforce explicit dual-modality alignment can learn to use context even when training data contains noisy textual signals. Future work includes reducing the cost of the verification procedure to generate a fully verified training set and developing methods to increase the diversity of generated contexts.

## Impact Statement

This paper presents work whose goal is to advance the field of Context-Aided Forecasting using automated methods. Considering that this task is one where human experts currently shine, improved automated methods would have an impact on the job market. Whether this impact will be positive due to more time series being amenable for accurate forecasting or negative due to fewer people needed to work on these forecasts is unknown to the authors.

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

# Appendix

## Table of Contents

## A. CAF-7M Dataset Details

### A.1. Construction Pipeline and Splits

We construct CAF-7M from the Chronos datasets collection[3] via a staged pipeline: (i) extract supervised forecasting windows from each source dataset, (ii) attach auxiliary fields (baseline difficulty proxies and textual context), and (iii) materialize temporally consistent in-domain splits together with fully held-out zero-shot test sets.

---

[3] https://huggingface.co/datasets/autogluon/chronos_datasets

*Table 4.* Train and Test split datasets used for CAF-7M.

| train | test |
|---|---|
| electricity_15min | dominick |
| m4_daily | ercot |
| m4_hourly | exchange_rate |
| m4_weekly | m4_quarterly |
| m4_monthly | m5 |
| mexico_city_bikes | monash_car_parts |
| monash_electricity_hourly | monash_australian_electricity |
| monash_electricity_weekly | monash_m1_yearly |
| monash_kdd_cup_2018 | monash_hospital |
| monash_london_smart_meters | monash_tourism_yearly |
| monash_pedestrian_counts | monash_tourism_quarterly |
| monash_rideshare | monash_tourism_monthly |
| monash_saugeenday | monash_m3_yearly |
| monash_temperature_rain | monash_cif_2016 |
| solar | monash_m1_monthly |
| solar_1h | m4_yearly |
| taxi_1h | monash_covid_deaths |
| taxi_30min | monash_m1_quarterly |
| uber_tlc_daily | monash_m3_monthly |
| uber_tlc_hourly | monash_weather |
| ushcn_daily | monash_m3_quarterly |
| weatherbench_daily | monash_fred_md |
| weatherbench_hourly_10m_u_component_of_wind | monash_nn5_weekly |
| weatherbench_hourly_10m_v_component_of_wind | monash_traffic |
| weatherbench_hourly_2m_temperature | nn5 |
| weatherbench_hourly_geopotential | |
| weatherbench_hourly_potential_vorticity | |
| weatherbench_hourly_relative_humidity | |
| weatherbench_hourly_specific_humidity | |
| weatherbench_hourly_temperature | |
| weatherbench_hourly_toa_incident_solar_radiation | |
| weatherbench_hourly_total_cloud_cover | |
| weatherbench_hourly_total_precipitation | |
| weatherbench_hourly_u_component_of_wind | |
| weatherbench_hourly_v_component_of_wind | |
| weatherbench_hourly_vorticity | |
| weatherbench_weekly | |
| wiki_daily_100k | |
| wind_farms_daily | |
| wind_farms_hourly | |

**Forecasting windows.** For each dataset, we generate rolling forecasting windows consisting of a history $\mathbf{z}_{1:P}$ and a future horizon $\mathbf{z}_{P+1:P+Q}$. History length $P$ is sampled uniformly from $[Q, 512]$, matching the maximum context length used by Chronos. The prediction length $Q$ is determined by dataset frequency metadata: 64 for hourly, daily, business-daily series, 5-minutely, and 15-minutely; 48 for 30-minutely; 52 for weekly; 12 for monthly; 8 for quarterly; and 5 for yearly.

**Window augmentation.** Each window is augmented with: (i) a baseline forecast and difficulty proxy (e.g., Chronos MASE) used for *hard/easy* stratification; and (ii) an LLM-generated scenario-style context. Context is produced by prompting `Llama-3-70B-Instruct` with dataset metadata and a serialized (history, future) pair using the template in Fig. 6. We use temperature $= 0.6$ and top_p $= 0.9$, following the recommended settings in the Hugging Face model card.[4] For scalability, we run the INT4-quantized checkpoint.[5]

**Dataset partitioning.** We partition the 65 source datasets into a training split of 40 datasets and a held-out test split of 25 datasets (see Table 4), following Chronos's *in-domain* and *zero-shot* evaluation protocol: the in-domain group is used for training, while the zero-shot group is excluded entirely from training and used only for evaluation. For in-domain datasets only, we compute dataset-specific temporal cutoffs from the empirical distribution of `start_idx` to avoid leakage. Specifically, for each in-domain dataset, we define a training cutoff at the 90th percentile and a validation cutoff at the 95th percentile of `start_idx`. These cutoffs are computed once and reused across all split generation steps. Using the cutoffs above, each in-domain dataset is split as:

- `indomain-train`: `start_idx` $\leq$ 90th percentile,
- `indomain-val`: 90th $<$ `start_idx` $\leq$ 95th percentile,
- `indomain-test`: `start_idx` $>$ 95th percentile.

We use `indomain-val` for training monitoring and `indomain-test` to evaluate in-domain fit; however, we make performance claims for DoubleCast only based on benchmarks on the held-out zero-shot datasets. We report DoubleCast results on `indomain-test` (HARD), `indomain-test` (EASY), and `indomain-test` (ALL). We define HARD and EASY using the Chronos MASE difficulty proxy: HARD if Chronos MASE $> 1.5$ and EASY otherwise, while ALL is the union of HARD and EASY.

For each zero-shot dataset, we construct a test split only. Because we do not train on these datasets, we do not enforce temporal cutoffs. Instead, we apply the same difficulty filtering described below to the full candidate pool, shuffle with a fixed seed, and sample to a fixed test budget. We report `zeroshot-test` (HARD), `zeroshot-test` (EASY), and `zeroshot-test` (ALL) (the union of HARD and EASY).

The primary splits of CAF-7M are `indomain-train`, which we called the training split of CAF-7M in the main text, and `zeroshot-test` (ALL), which we called the testing split of CAF-7M in the main text.

**DP-based context verification for evaluation splits.** To ensure that textual context provides *measurable* predictive utility, we construct a *verified* pool on evaluation splits via DP verification. For each candidate window, we query a downstream probabilistic forecaster twice—once with the generated context and once without—compute CRPS for both, and retain the window if the context improves CRPS. When drawing from this verified pool, we enforce the DP constraint alongside any temporal and/or difficulty constraints above. All evaluation splits in this work are produced with DP verification enabled. For `indomain-val` and `indomain-test`, we use `Qwen2.5-7B-Instruct` as the DP verifier to reduce cost. For `zeroshot-test`, we use `GPT-5.2` to maximize verification fidelity, yielding a higher-quality test set for assessing context complementarity.

Although forecaster-based filtering improves context quality, we do not apply it to the full training split because of its computational cost. In a smaller in-domain validation experiment, filtered data was more sample-efficient at matched budgets, while larger unfiltered sets partially recovered the gap. Table 5 reports validation cross-entropy after roughly 40k training steps.

These results support a quantity–quality trade-off: filtering can improve sample efficiency, but the scale of the unfiltered corpus remains valuable. Because this experiment used a cheaper filter than the `GPT-5.2`-based zero-shot test verification, we treat it as suggestive rather than definitive.

---

[4] https://huggingface.co/meta-llama/Meta-Llama-3-70B-Instruct
[5] https://huggingface.co/RedHatAI/Llama-3.3-70B-Instruct-quantized.w4a16

*Table 5.* In-domain validation cross-entropy after roughly 40k training steps for models trained on filtered or unfiltered subsets. Lower is better.

| Budget | Filtered | Unfiltered | Gap |
|--------|----------|------------|------|
| 50k | $\sim$2.89 | $\sim$2.95 | +2.0% |
| 25k | $\sim$3.10 | $\sim$3.24 | +4.3% |
| 10k | $\sim$3.63 | $\sim$3.75 | +3.2% |
| 5k | $\sim$4.11 | $\sim$4.20 | +2.1% |

**Human evaluation of generated contexts.** As a complementary surface-quality check, we asked 11 annotators to evaluate 30 generated context examples. Each example showed the historical portion of a time series, the future portion, and the generated context. Annotators answered whether (i) the background made clear what quantity was plotted and (ii) the scenario was plausible given the background. We aggregate binary responses by majority vote across annotators.

*Table 6.* Human evaluation of generated context quality over 30 examples. We report the percentage of examples with positive majority-vote judgments.

| Criterion | Positive examples |
|-----------|-------------------|
| Clear plotted quantity | 80% |
| Plausible scenario | 82% |

This evaluation suggests that most generated contexts are understandable and plausible to human annotators. However, it is not our primary quality criterion: contexts may still contain hallucinations or surface-level plausibility issues, so CAF-7M relies on forecast improvement under the DP verifier as its main acceptance signal.

**Semantic validity pre-filter (zero-shot only).** Because DP verification with `GPT-5.2` is expensive, we apply a lightweight semantic validity check before DP verification on zero-shot candidates using an LLM judge. Given a generated context, the judge answers two binary questions: (Q1) whether the context states what variable the time series represents, and (Q2) whether the described scenario/event is plausibly related to that variable (Fig. 7). This judge stage is used only as a coarse pre-filter to remove clearly invalid contexts; DP verification remains the primary acceptance mechanism.

**Global splits and sampling.** After generating per-dataset splits, we concatenate windows across all in-domain datasets to form global `indomain-train`, `indomain-val`, and `indomain-test` splits, and concatenate all zero-shot datasets to form a global `zeroshot-test` split. When reporting stratified evaluation variants (HARD/EASY/ALL), we sample windows from the corresponding candidate pools using a fixed seed and a fixed per-dataset or global budget.

**Training subsampling for scaling studies.** Finally, we create subsampled variants of the in-domain training set (e.g., 0.1%, 1%, 10%, 25%, 50%, etc.) by shuffling training indices with a fixed seed and selecting the first $n$ windows.

### A.2. Dataset Statistics

In this subsection, we summarize the composition and difficulty characteristics of the CAF-7M splits through dataset-level statistics. Specifically, we report per-dataset window counts, the distributions of history and forecast-horizon lengths (both overall and stratified by dataset), and the baseline difficulty proxy Chronos MASE stratified by dataset. Dataset count bar charts sort datasets by window count in descending order. Length distributions are visualized using histograms and box-and-whisker plots. Boxplots follow Matplotlib defaults: boxes show the interquartile range with the median indicated, whiskers extend to $1.5 \times \text{IQR}$, and outliers are not shown. For all MASE visualizations, we exclude non-finite values and values above 5 prior to plotting.

```
"
You are working with the following dataset:

{domain_hint}

Your task is to generate textual context that aids in forecasting the future of a time series from this dataset. The
     generated context should introduce information that helps explain the observed changes but is not directly
     obvious from the past numerical values. Below are examples of the style of contexts you should generate:

1. Background: This dataset represents electricity consumption, measured in kilowatts (kW), in City A. Scenario:
     Suppose a heat wave occurs in City A from 2013-05-28 12:00:00 to 2013-05-28 14:00:00, resulting in excessive
     use of air conditioning and electricity consumption increasing to four times the usual level.
2. Constraint: In the forecast, the values are assumed to be bounded above by 6.29.

These examples illustrate the three key types of contextual information:

**Background**: Historical details that may influence future values.
**Scenario**: Plausible events that could lead to changes in future values.
**Constraint**: Assumptions or bounds that restrict the possible range of future values.

Your generated context should include one or more of these types. Now, using the following time series {target_var_
     text}from this dataset, generate a plausible textual context to help forecast. The data is in (timestamp, value
     ) format, with the forecast horizon starting at {window['future_timestamp'][0]}.

Historical Time Series:
<past_target>
{history}
</past_target>

Ground Truth for Future:
<future_target>
{future}
</future_target>

Follow these guidelines when generating the context:

- **Focus** on non-stationary segments in the forecast horizon---such as trend shifts (e.g., from fluctuating to a
     steady decline) or non-recurring patterns (e.g., unusually low peaks). A change is major if it shows a
     sustained, non-seasonal deviation (e.g., surge, drop, plateau) lasting at least one full seasonal cycle. Ignore
      minor fluctuations lasting only one or two timestamps.

- **If no major changes are present**, generate only a **Background** summary describing historical trends and
     seasonality, and how they are expected to persist.

- **If major changes are observed** (in all or part of the forecast):
- If multiple such changes occur, describe **only the most impactful one**, based on magnitude or duration.
- Create a realistic and specific **Scenario** describing a plausible causal event. If several plausible causes
     exist, choose the most likely one. Avoid vague or speculative reasoning.
- Include a **Constraint** only if forecast values are clearly and consistently bounded (e.g., all values equal to
     6.29). Omit this part entirely if no constraint is observed. Do not write "Constraint: None" or similar.
- Ensure the scenario is consistent with both the **domain description** and the patterns in <future_target>.
- Describe the impact on future values using **relative terms** (e.g., "2x" higher", "20% lower") rather than exact
      values.
- If the scenario affects only part of the forecast, clearly specify the **start and end times** using the format
     YYYY-MM-DD HH:MM:SS.
Use HH:MM:SS only if all future timestamps are on the same day.
- You may refer to **historical timestamps** only if they clearly relate to recurring or causal patterns.
- Include a **short summary** in <short_desc> (e.g., "traffic jam", "holiday closure").
If no scenario is generated, return: <short_desc></short_desc>.

{variety_instruction}

- **Keep the context concise** --- ideally **150-250 words**. Avoid unnecessary elaboration or repetition.

- **Do not include your reasoning steps** or explain how the context was inferred. Output only the final description
      in a **factual and declarative** style.

Output Format:

<context_abs>put context here</context_abs>
<short_desc>put short description here if applicable</short_desc>
```

*Figure 6.* Template of the prompt sent to `Llama-3-70B-Instruct` for the context generation. {`domain_hint`} is filled with the dataset metadata; {`history`} and {`future`} are the serialized time series data; and {`variety_instruction`} contains a resume of the previously-generated context with instructions to not repeat it.

```
 "
Given the following context information about a time series:

{context}

Answer these two questions with ONLY "yes" or "no":

Q1: Does the context state what variable the time series represents (e.g., profit, demand, load), even if units or
    aggregation level are not specified?
Q2: Is the scenario/event related to the quantity being measured in the time series (i.e., it describes events that
    would reasonably affect that specific variable)?

Format your answer as:
Q1: [yes/no]
Q2: [yes/no]
```

*Figure 7.* Template of the prompt sent to `GPT-5.2` to judge the semantic validity of a window context. {`context`} is filled with the generated context.

**indomain-train.**

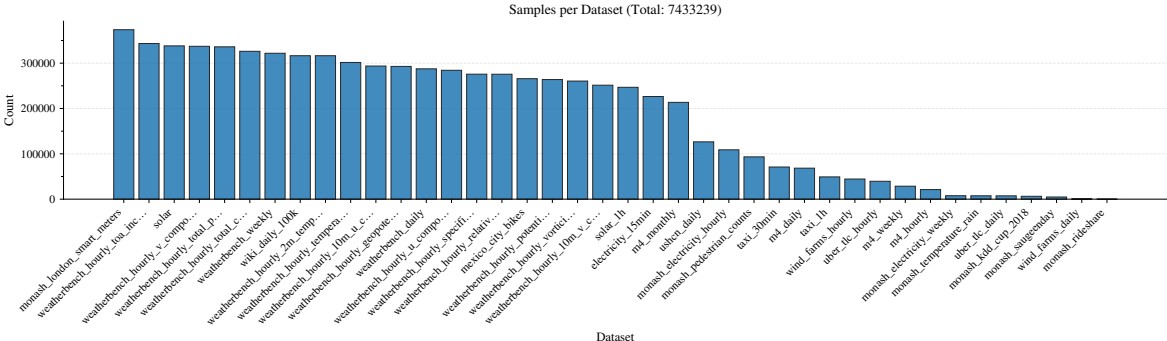

*Figure 8.* Per-dataset window counts for `indomain-train`.

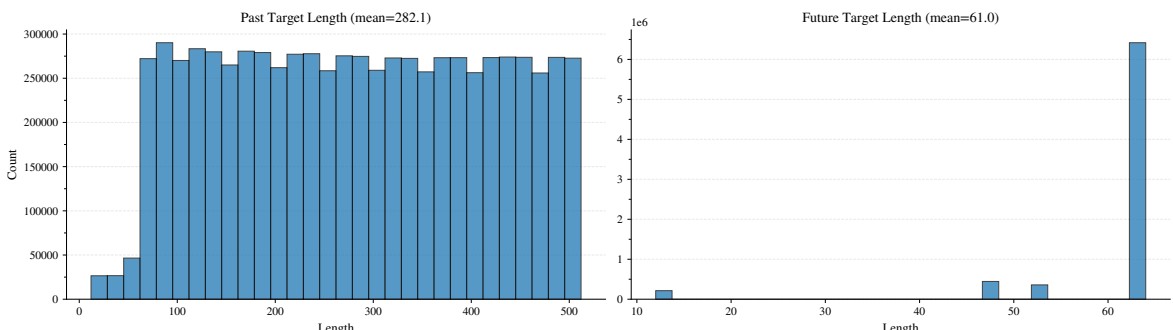

*Figure 9.* Overall distribution of history and forecast-horizon lengths for `indomain-train`.

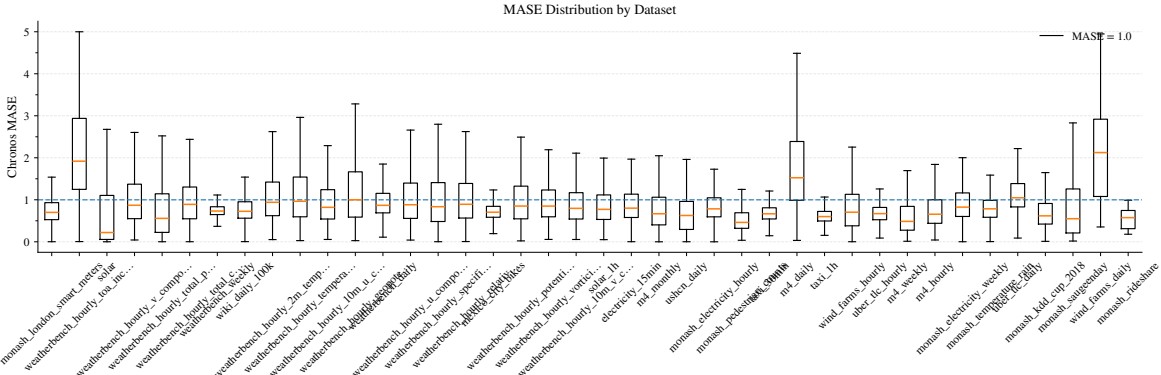

*Figure 10.* Baseline difficulty proxy (Chronos MASE) by dataset for `indomain-train`.

**indomain-test** (ALL).

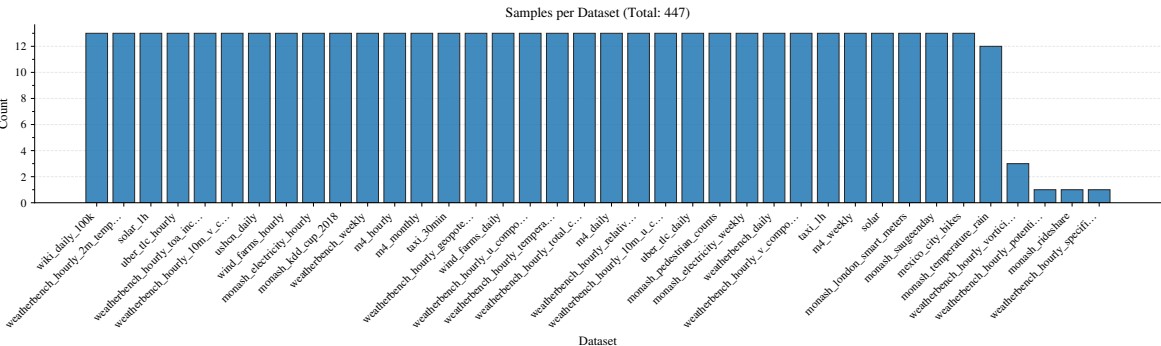

*Figure 11.* Per-dataset window counts for `indomain-test` (ALL).

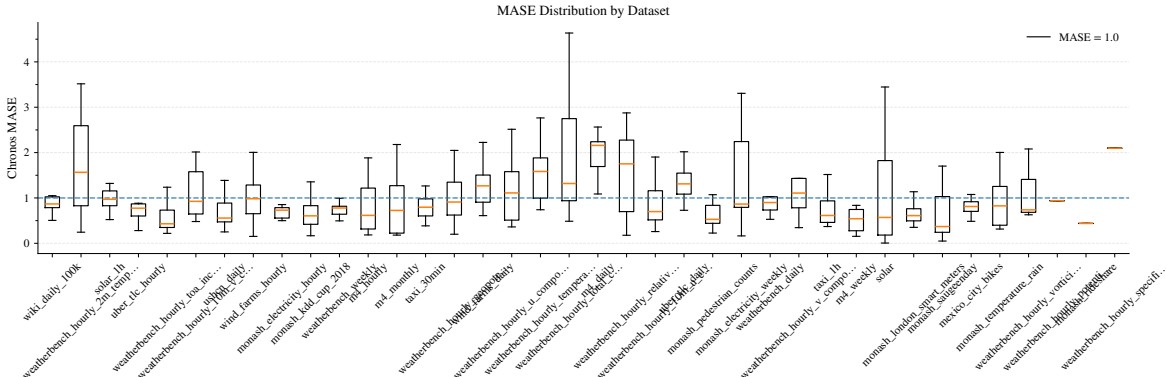

*Figure 12.* Baseline difficulty proxy (Chronos MASE) by dataset for `indomain-test` (ALL).

**`indomain-test`** (HARD).

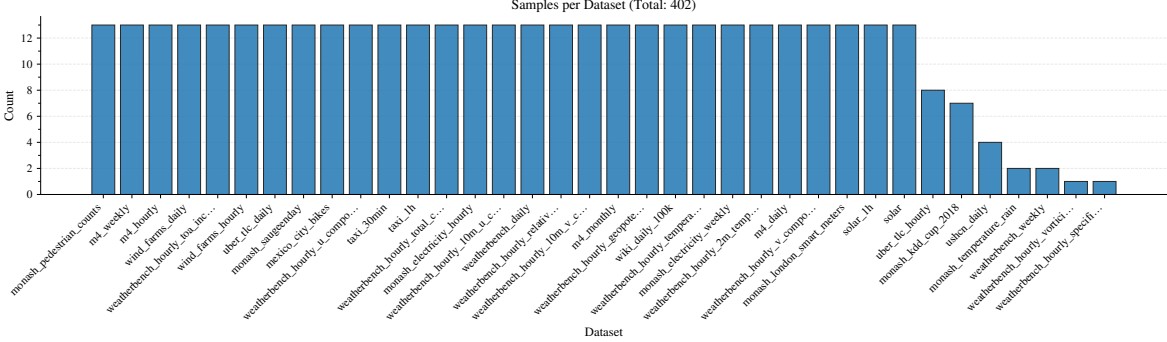

*Figure 13.* Per-dataset window counts for `indomain-test` (HARD).

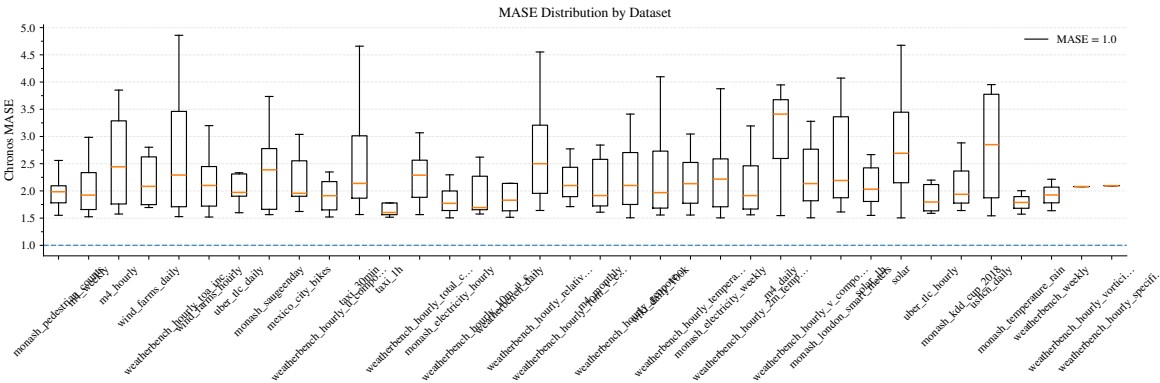

*Figure 14.* Baseline difficulty proxy (Chronos MASE) by dataset for `indomain-test` (HARD).

**`indomain-test` (EASY).**

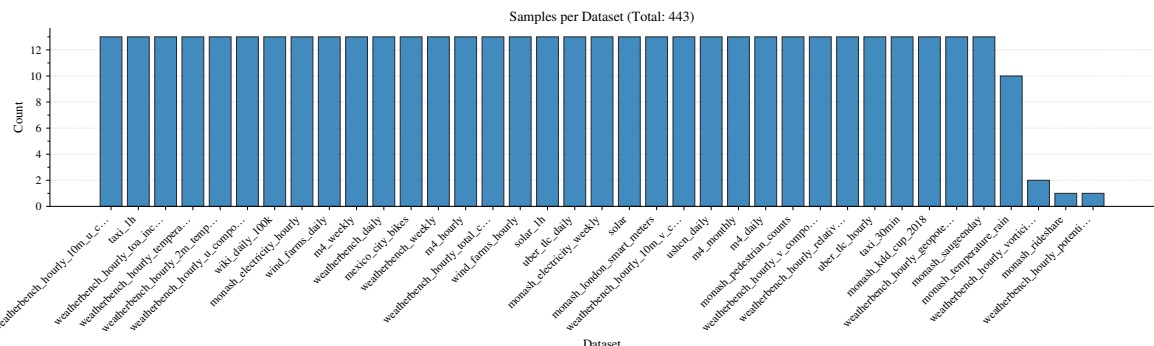

*Figure 15.* Per-dataset window counts for `indomain-test` (EASY).

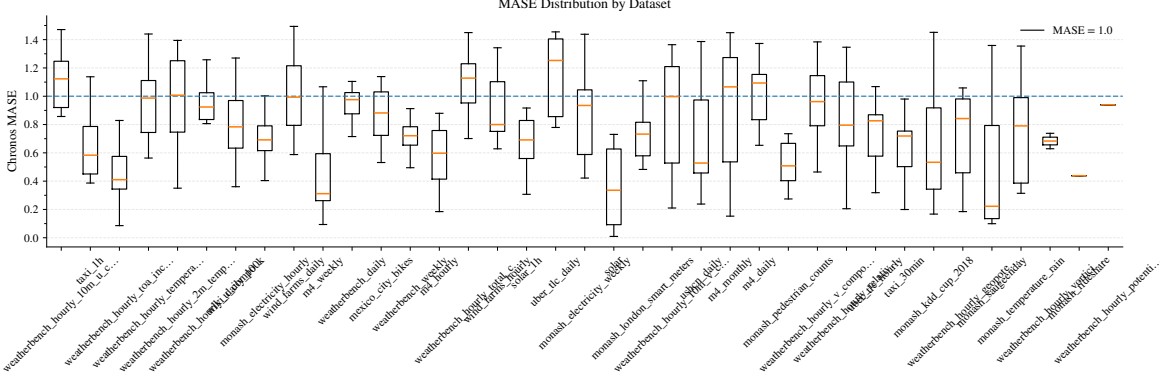

*Figure 16.* Baseline difficulty proxy (Chronos MASE) by dataset for `indomain-test` (EASY).

**`zeroshot-test` (HARD).**

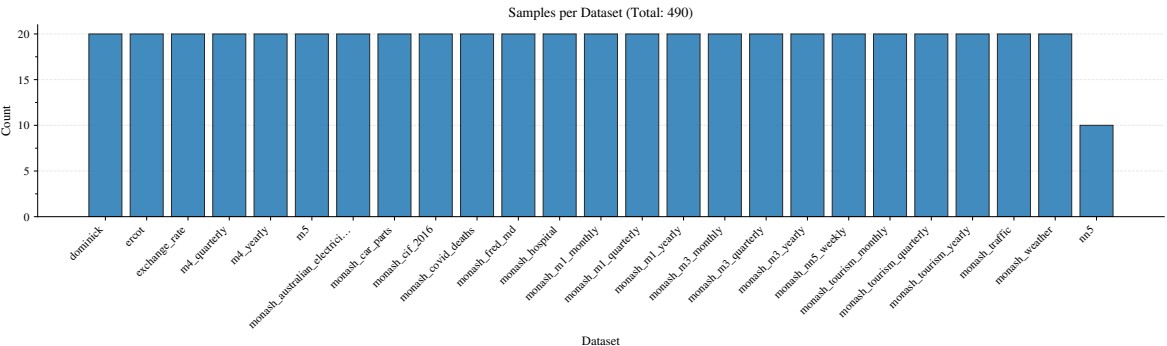

*Figure 17.* Per-dataset window counts for `zeroshot-test` (HARD).

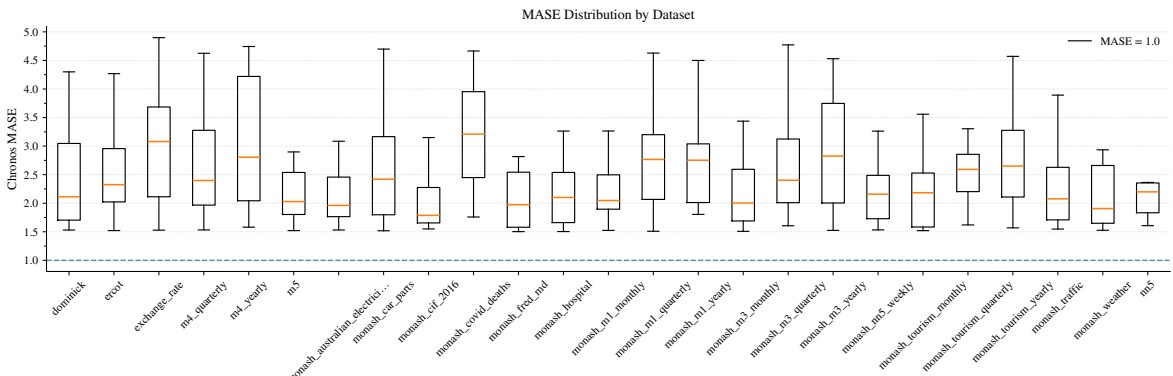

*Figure 18.* Baseline difficulty proxy (Chronos MASE) by dataset for `zeroshot-test` (HARD).

**`zeroshot-test` (EASY).**

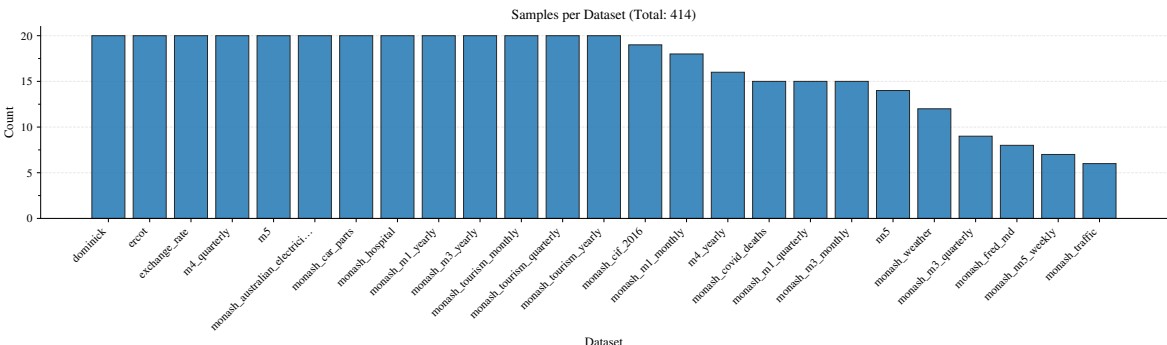

*Figure 19.* Per-dataset window counts for `zeroshot-test` (EASY).

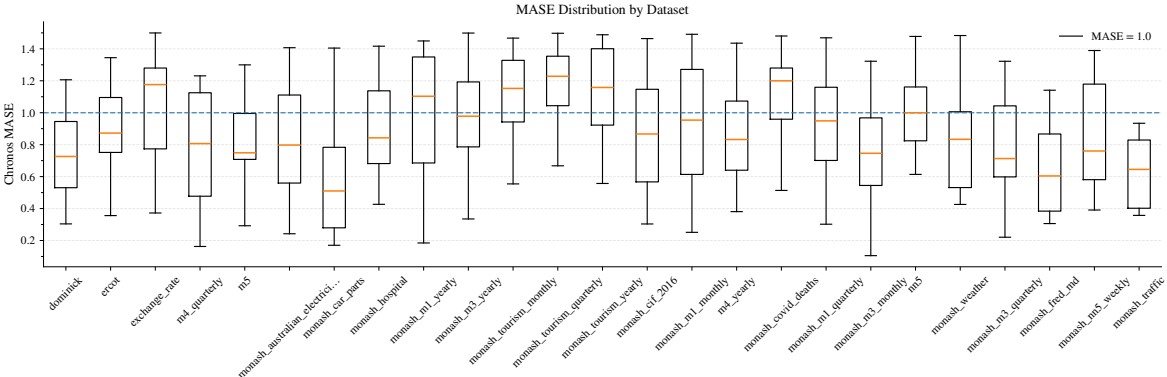

*Figure 20.* Baseline difficulty proxy (Chronos MASE) by dataset for `zeroshot-test` (EASY).

## B. DoubleCast Architecture Details

### B.1. DualT5 Decoder and Context Injection

DoubleCast augments the Chronos (T5) decoder with an additional text encoder–decoder cross-attention sublayer, enabling autoregressive forecasting conditioned on both numerical history and unstructured textual context. For a given forecasting window, the Chronos time-series encoder maps the historical series into encoder states $e_{\text{ts}} \in \mathbb{R}^{L_{\text{ts}} \times d_{\text{model}}}$. In parallel, a pretrained text encoder maps the input context string into hidden states $e_{\text{ctx}} \in \mathbb{R}^{L_{\text{ctx}} \times d_{\text{ctx}}}$. These same $e_{\text{ts}}$ and $e_{\text{ctx}}$ are provided to every (dual) decoder block, as shown in Fig. 2.

**Block structure and ordering.** A `DualT5` decoder block consists of the following sublayers in sequence: (i) masked self-attention, (ii) standard Chronos encoder–decoder cross-attention layer, over $e_{\text{ts}}$, (iii) DualT5 cross-attention layer, over $e_{\text{ctx}}$, and (iv) a feed-forward network (FFN). Each sublayer uses the standard T5 pre-norm convention: layer normalization is applied to the sublayer input, the transformation is applied, and the result is added back via a residual connection.

More concretely, within each decoder block, the intermediate hidden state $\tilde{h}^{(0)}$ produced by the self-attention layer serves as the query for cross-attention over the time-series encoder outputs $e_{\text{ts}}$, yielding $\tilde{h}^{(1)}$. This state then becomes the query for a secondary cross-attention mechanism over the projected textual context embeddings $e_{\text{ctx}}$. These embeddings are drawn from an intermediate layer of the pretrained LLM architecture, since these often capture rich semantic representations useful for downstream tasks (Skean et al., 2025).

To accommodate differing hidden dimensions between the text encoder and Chronos decoder, we implement cross-attention via an adapter layer: keys and values are generated by learned projections $W_K, W_V \in \mathbb{R}^{d_{\text{ctx}} \times d_{\text{ts}}}$, which map the text-encoder outputs (dimension $d_{\text{ctx}}$) into the decoder's attention space (dimension $d_{\text{ts}}$). This adapter-style design enables seamless integration of arbitrary LLMs without altering their native architectures.

Let $h_{\text{in}}$ denote the input hidden state to a decoder block. The sequential updates can be written as

$$\tilde{h}^{(0)} = h_{\text{in}} + \text{SelfAttn}\big(\text{LN}(h_{\text{in}})\big), \tag{4}$$

$$\tilde{h}^{(1)} = \tilde{h}^{(0)} + \text{CrossAttn}_{\text{ts}}\big(\text{LN}(\tilde{h}^{(0)}), e_{\text{ts}}\big), \tag{5}$$

$$\tilde{h}^{(2)} = \tilde{h}^{(1)} + \text{CrossAttn}_{\text{ctx}}\big(\text{LN}(\tilde{h}^{(1)}), e_{\text{ctx}}\big), \tag{6}$$

$$h_{\text{out}} = \tilde{h}^{(2)} + \text{FFN}\big(\text{LN}(\tilde{h}^{(2)})\big), \tag{7}$$

where $h_{\text{in}}$ denotes the input hidden state to the decoder block (either the output of the preceding block or the initial token embeddings); $\tilde{h}^{(i)}$ for $i = 0, 1, 2$ are the intermediate hidden states; and $h_{\text{out}}$ is the final hidden state of the decoder block.

where $\text{CrossAttn}_{\text{ts}}$ is the original Chronos encoder–decoder attention and $\text{CrossAttn}_{\text{ctx}}$ is the added text-conditioning attention.

**Dual block placement.** DoubleCast supports applying text-conditioning in all decoder layers or restricting it to a specified subset of decoder layers. Layers not selected remain standard T5 blocks and thus do not attend to $e_{\mathrm{ctx}}$. This provides a simple architectural knob to modulate where and how strongly text influences generation.

### B.2. Text Encoder and Adapter Projections

**Text encoder and feature selection.** DoubleCast employs an independent pretrained text encoder to compute contextual embeddings. The model extracts textual representations from a configurable hidden layer index `text_encoder_layer_index` (default: second-to-last, $-2$), yielding $e_{\mathrm{ctx}} \in \mathbb{R}^{L_{\mathrm{ctx}} \times d_{\mathrm{ctx}}}$.

**Adapter-style key/value projections.** To accommodate dimensional mismatch between the text encoder width $d_{\mathrm{ctx}}$ and the Chronos decoder attention space, DoubleCast implements the text cross-attention via adapter projections on keys and values. Concretely, the added text attention reuses the standard T5 query projection from decoder states, but replaces the key/value projections with learned linear maps

$$\boldsymbol{K}_{\mathrm{ctx}} = \boldsymbol{e}_{\mathrm{ctx}} W_K, \qquad \boldsymbol{V}_{\mathrm{ctx}} = \boldsymbol{e}_{\mathrm{ctx}} W_V, \tag{8}$$

where $W_K, W_V \in \mathbb{R}^{d_{\mathrm{ctx}} \times d_{\mathrm{attn}}}$ map into the decoder attention inner dimension $d_{\mathrm{attn}} = h \cdot d_{kv}$. This design enables seamless integration of arbitrary pretrained text encoders without modifying their native architectures.

**Initialization and parameter-efficient tuning.** The overall model is initialized from a pretrained Chronos checkpoint: all compatible encoder/decoder weights are copied, and only the newly introduced text cross-attention parameters are randomly initialized following T5 initialization. In our training setup, we freeze the entire backbone by default and selectively unfreeze (i) the text cross-attention parameters (including its layer norm) and (ii) the FFN sublayer within the dual blocks, yielding a parameter-efficient adaptation focused on learning how to condition Chronos forecasts on text.

### B.3. Design Variants

**Architectural variants.** DoubleCast admits natural variants along three axes: (i) the choice of text encoder, (ii) the depth from which $e_{\mathrm{ctx}}$ is extracted (`text_encoder_layer_index`), and (iii) the set of decoder layers equipped with text cross-attention (`dual_block_placement`). In this work, we fix these degrees of freedom (Qwen3-14B, second-to-last layer, and text-conditioning in all decoder layers) to focus ablations on context quality and dataset scale rather than exhaustive architectural search.

## C. Training and Evaluation Protocols

### C.1. Training and Inference Settings

**Implementation and reproducibility.** All experiments are implemented in `PyTorch` using `HuggingFace Transformers` training utilities. We fix the random seed to $42$ for Python and NumPy/`torch` RNGs.

**Model initialization.** We initialize DoubleCast from a pretrained Chronos backbone (default: `amazon/chronos-t5-large`[6]) and a pretrained text encoder (e.g., `Qwen/Qwen3-14B`[7]). Text features are extracted from a specified hidden layer of the text encoder (default: second-to-last). The dual cross-attention modules are injected into a configurable subset of decoder blocks; unless otherwise noted, we enable them for all decoder blocks.

**Trainable parameters.** To stabilize optimization and isolate the effect of context fusion, we freeze all pretrained parameters and only fine-tune the newly introduced (or context-facing) components within each `DualT5DecoderBlock`. Concretely, for each enabled dual block we unfreeze (i) the cross-attention parameters (`EncDecAttention`), (ii) the corresponding layer-norm parameters, and (iii) the block feed-forward network (FFN). All other parameters remain frozen throughout training.

**Optimization and training setup.** We train with AdamW (`adamw_torch_fused`) using a cosine learning-rate schedule and weight decay. The global learning rate is $10^{-4}$ with warmup ratio $0.005$ and weight decay $0.01$, and we apply gradient clipping with max norm $1.0$. Training runs for a fixed number of optimizer steps (`max_steps=400,000`) with gradient accumulation (`gradient_accumulation_steps=2`) and per-device batch size (`per_device_train_batch_size=8`).

---

[6] https://huggingface.co/amazon/chronos-t5-large
[7] https://huggingface.co/Qwen/Qwen3-14B

We save checkpoints every `save_steps` and evaluate every `eval_steps` on all validation datasets. Unless otherwise noted, we report results from the final checkpoint at 400,000 steps. To regularize the fusion pathway, we optionally down-weight updates to FFN parameters inside the dual blocks by applying a multiplicative learning-rate factor (default 0.01) via separate optimizer parameter groups. Training is conducted on a single NVIDIA H100 (80 GB) GPU using `bfloat16` weights and activations.

**Fine-tuning on CGTSF (ChatTime).** To adapt DoubleCast to CGTSF, we fine-tune the pretrained DoubleCast on the training split of CGTSF dataset. Using the checkpoint of DoubleCast trained in CAF-7M, we fine-tune for additional 50,000 steps using AdamW with a cosine schedule, learning rate $10^{-6}$, warmup ratio 0.005, weight decay 0.01, and gradient clipping at max norm 1.0. We use per-device batch size 8 with gradient accumulation 2 (effective batch size 16). Similarly, during fine-tuning, we train only the text cross-attention, layer-norm, and FFN in each decoder block while keeping others frozen. In contrast to pretraining, we set the FFN learning-rate multiplier to 0.5 to allow more rapid adaptation of the decoder feed-forward sublayers to the target context style.

**Prompt formatting for textual context.** We serialize the textual context (and optional metadata such as forecast start date, frequency, and scale) into a single string that is tokenized by the text encoder. We consider two templates: a *naive* template that passes the raw context, and a *structured* template that wraps metadata and context in explicit tags, as illustrated in Fig. 21. Unless otherwise stated, DoubleCast uses the structured template in all experiments.

```
"
<info>
forecast_start_date=2025-06-25 00:00:00
frequency=D
scale_factor=0.1234
</info>

<context>
Background: The time series exhibits seasonal fluctuations...
</context>
```

*Figure 21.* Text encoder prompt.

**Inference and test evaluation.** At inference time, each evaluation instance provides a numerical history `past_target` and a textual context string. The forecast horizon $T$ is set to the instance-specific future length, i.e., $T = |\texttt{future\_target}|$.

DoubleCast produces probabilistic forecasts via ancestral sampling from the decoder with `num_samples` Monte Carlo trajectories. We use the same generation procedure as Chronos—sampling with temperature and nucleus/top-$k$ filtering as specified by the Chronos configuration—and generate sequences in chunks when $T$ exceeds the model's native prediction length. Concretely, the pipeline iteratively predicts up to `prediction_length` steps per call and, when additional steps are required, appends the per-step sample median to the numerical context before continuing, ensuring a consistent autoregressive rollout while retaining stochasticity within each chunk. For comparability and efficiency, we run inference in `bfloat16` on GPU

We report CRPS (and standard error across windows) using the same empirical-sample estimator as in Appendix C.2. Finally, when Chronos reference forecasts are available for the same split, we compute the per-window CRPS win-rate of DoubleCast against Chronos under identical evaluation conditions.

### C.2. Metrics

**Notation.** For an evaluation window with horizon length $T$, let the ground truth be $y_{1:T} \in \mathbb{R}^T$ and the probabilistic forecast be represented by $N$ Monte Carlo samples $\{x_{1:T}^{(n)}\}_{n=1}^N$.

**CRPS.** We use the Continuous Ranked Probability Score (CRPS) as the primary probabilistic metric, following the standard empirical-sample formulation used in Williams et al. (2025). For each horizon step $t$, CRPS can be written as

$$\mathrm{CRPS}(X_t, y_t) = \mathbb{E}|X_t - y_t| - \tfrac{1}{2}\mathbb{E}|X_t - X_t'|, \tag{9}$$

where $X_t$ and $X_t'$ are i.i.d. draws from the empirical distribution defined by the $N$ samples at step $t$. We report the horizon-averaged CRPS

$$\mathrm{CRPS}_{\mathrm{mean}}(X, y) = \frac{1}{T}\sum_{t=1}^{T}\mathrm{CRPS}(X_t, y_t). \tag{10}$$

To compare across heterogeneous series scales, we normalize by the mean absolute target magnitude

$$s(y) = \frac{1}{T} \sum_{t=1}^{T} |y_t|, \qquad \text{nCRPS}(X, y) = \frac{\text{CRPS}_{\text{mean}}(X, y)}{s(y) + \varepsilon}, \tag{11}$$

with $\varepsilon = 10^{-6}$ for numerical stability. As in our evaluation pipeline, we cap per-window scores at $c = 5$:

$$\widetilde{\text{nCRPS}}(X, y) = \min\{\text{nCRPS}(X, y), c\}. \tag{12}$$

We then aggregate over a split by the simple mean (all instance weights are 1):

$$\overline{\text{nCRPS}} = \frac{1}{M} \sum_{i=1}^{M} \widetilde{\text{nCRPS}}_i. \tag{13}$$

**Win-rate.** Win-rate is computed against Chronos on a per-window basis using the same (normalized, clipped) CRPS. Let $m_i$ be the model score and $b_i$ the Chronos score on window $i$. The win-rate is

$$\text{WinRate}(m \prec b) = \frac{1}{M} \sum_{i=1}^{M} \mathbb{I}[m_i < b_i], \tag{14}$$

where ties are counted in the denominator but not as wins.

**MAE.** For CGTSF we follow ChatTime and report normalized MAE. We form a point forecast via the per-timestep sample median,

$$\hat{x}_t = \text{median}\{x_t^{(n)}\}_{n=1}^{N}, \tag{15}$$

and compute

$$\text{MAE}(X, y) = \frac{1}{T} \sum_{t=1}^{T} |\hat{x}_t - y_t|, \qquad \text{nMAE}(X, y) = \frac{\text{MAE}(X, y)}{s(y) + \varepsilon}, \tag{16}$$

with the same scale $s(y)$ as in Eq. (11). We apply the same per-window cap $c = 5$ and report the mean over a split:

$$\overline{\text{nMAE}} = \frac{1}{M} \sum_{i=1}^{M} \min\{\text{nMAE}_i, c\}. \tag{17}$$

**MASE.** We also compute the Mean Absolute Scaled Error (MASE) to characterize per-window forecast difficulty. For a window with horizon length $T$, let $\hat{y}_{1:T}$ be the Chronos point forecast and $y_{1:T}$ the ground truth. Let $z_{1:L}$ denote the historical context used for conditioning. Given a seasonal period $S$, we define the scaling term

$$d(z; S) = \begin{cases} \frac{1}{L-S} \sum_{t=S+1}^{L} |z_t - z_{t-S}|, & L > S, \\ \frac{1}{L-1} \sum_{t=2}^{L} |z_t - z_{t-1}|, & L \leq S, \end{cases} \tag{18}$$

i.e., the mean absolute error of a seasonal naive forecast on the history, with a non-seasonal fallback when insufficient history is available. The window-level MASE is then

$$\text{MASE}(\hat{y}, y; z, S) = \begin{cases} \frac{\frac{1}{T} \sum_{t=1}^{T} |\hat{y}_t - y_t|}{d(z; S)}, & d(z; S) > 0, \\ 0, & d(z; S) = 0 \ \wedge \ \frac{1}{T} \sum_{t=1}^{T} |\hat{y}_t - y_t| = 0, \\ +\infty, & d(z; S) = 0 \ \wedge \ \frac{1}{T} \sum_{t=1}^{T} |\hat{y}_t - y_t| > 0. \end{cases} \tag{19}$$

We select $S$ deterministically from the dataset frequency metadata (e.g., $S=24$ for hourly, $S=7$ for daily, $S=12$ for monthly; otherwise $S=1$).

# D. Extended Related Work

Following Zhang et al. (2025), we distinguish between *alignment-based* and *prompt-based* methods for context-aided forecasting.

## D.1. Alignment-Based Methods

Among the methods that align time series and textual modalities via dedicated architecture, UniTime (Liu et al., 2024b) concatenates time-series and textual embeddings, and feeds the resulting vector to a language-TS transformer based on GPT2. Time-LLM (Jin et al., 2024) introduces patch reprogramming to map time series into textual prompts and adds as prefix domain description, instructions and statistics of the time-series, leveraging frozen LLMs to perform forecasting with minimal training. Dual-forecaster (Wu et al., 2025) combines the time-series modality with textual descriptions of its history and its future via attention mechanisms. Time-VLM (Zhong et al., 2025) broadens the modality set by coupling vision-style encodings of temporal patterns with generated textual descriptions, then leveraging a frozen vision-language backbone to provide augmented representations that particularly help in few/zero-shot regimes.

## D.2. Prompt-Based Methods

Another line of work prompt LLMs with both numerical and contextual inputs to generate probabilistic forecasts (Gruver et al., 2023; Requeima et al., 2024; Williams et al., 2025). Subsequent work tightens the alignment between language representations and time-series structure. $S^2$IP-LLM (Pan et al., 2024) learns prompts in a joint semantic space that aligns pre-trained LLM embeddings with time-series features, improving transfer while reducing heavy fine-tuning. In parallel, AutoTimes (Liu et al., 2024c) repurposes decoder-only LLMs as autoregressive forecasters by projecting time series into the language token space; Wang et al. (2024) use LLM agents to retrieve, filter, and align news events with time-series fluctuations, iteratively refining forecasts through event-aware reasoning. Tan et al. (2024) find that for several recent LLM-for-forecasting models, removing or replacing the LLM with simple attention often maintains or improves accuracy, suggesting that LLMs help most when they bring truly complementary knowledge, e.g., coming from textual sources. This question is further studied in Zhang et al. (2025), where the authors empirically identify the conditions by which textual context helps improve forecasting performance. In particular, they find that textual context is most beneficial when it conveys information not inferable from time series, such as domain metadata or future events, and that architectures which explicitly align the two modalities are more reliable than treating series purely as text.

# E. Baselines

## E.1. Chronos

We use the publicly available implementation of Chronos (Ansari et al., 2024) from `https://github.com/amazon-science/chronos-forecasting`. In this work, we report results using `chronos-large` and run all Chronos inference on a single H100 GPU.

## E.2. ChatTime

We evaluate ChatTime in the zero-shot setting using the released `ChatTime-1-7B-Chat` checkpoint (`https://huggingface.co/ChengsenWang/ChatTime-1-7B-Chat`), following the evaluation procedure provided in the authors' repository (`https://github.com/ForestsKing/ChatTime`).

## E.3. Time-LLM

We implement the Time-LLM (Jin et al., 2024) architecture using the authors' official codebase (`https://github.com/KimMeen/Time-LLM`). For zero-shot evaluation, we use the publicly released checkpoints and settings reported by Williams et al. (2025). For in-domain adaptation on CAF-7M, we further train Time-LLM on our `indomain-train` split. Concretely, we train for $400,000$ optimization steps with BF16 mixed precision on a single H100 GPU, using batch size $8$ with gradient accumulation $2$, AdamW with learning rate $10^{-4}$, warmup ratio $0.005$, weight decay $0.01$, and gradient clipping at $1.0$. We use a forecasting setup with sequence length $128$, label length $64$, and prediction length $64$, and adopt the default long-term forecasting task configuration. Unless otherwise stated, we train a lightweight projection head with $d_{\text{model}} = 32$ and $d_{\text{ff}} = 128$, and optimize for MAE. Similarly, the final-checkpoint was used for experiments in this paper.

### E.4. Direct Prompt (DP)

Following prior work on prompt-based contextual forecasting (Williams et al., 2025), we evaluate DP baselines that queries an instruction-tuned LLM to produce the full forecast in a single pass. Given an instance-level textual context, a history window formatted as (timestamp, value) pairs, and a list of prediction timestamps, the model is prompted to output the forecast in the same (timestamp, value) format inside `<forecast>` tags. We use fixed prompt templates (with and without context) and refer to Figs. 22 and 23 for the exact prompts. Following the methodology in (Williams et al., 2025), we sample 25 independently generated forecasts per instance.

```
"
I have a time series forecasting task for you.

Here is some context about the task. Make sure to factor in any background knowledge,
satisfy any constraints, and respect any scenarios.
<context>
{context}
</context>

Here is a historical time series in (timestamp, value) format:
<history>
{past_time}
</history>

Now please predict the value at the following timestamps: {future_time_index_concat}.

Return the forecast in (timestamp, value) format in between <forecast> and </forecast> tags.
Do not include any other information (e.g., comments) in the forecast.

Example:
<history>
(t1, v1)
(t2, v2)
(t3, v3)
</history>
<forecast>
(t4, v4)
(t5, v5)
</forecast>
```

*Figure 22.* Template of the prompt for the Direct Prompt forecasting method. {`context`} is filled with the entry context; {`past_time`} is the serialized time series data from the history window; and {`future_time_index_concat`} contains the timestamps at which to do the forecast.

```
"
I have a time series forecasting task for you.

Here is a historical time series in (timestamp, value) format:
<history>
{past_time}
</history>

Now please predict the value at the following timestamps: {future_time_index_concat}.

Return the forecast in (timestamp, value) format in between <forecast> and </forecast> tags.
Do not include any other information (e.g., comments) in the forecast.

Example:
<history>
(t1, v1)
(t2, v2)
(t3, v3)
</history>
<forecast>
(t4, v4)
(t5, v5)
</forecast>
```

*Figure 23.* Template of the prompt for the Direct Prompt forecasting method when called without the context. {`past_time`} is the serialized time series data from the history window and {`future_time_index_concat`} contains the timestamps at which to do the forecast.

# F. Additional Results and Analyses

## F.1. Context Complementarity in the CAF-7M

Fig. 4 reports mean CRPS (left) and win-rate versus Chronos (right) on the EASY split, using the same accepted/rejected partition and the same three context settings (correct context, no context, swapped context). The overall trends match the HARD split but are generally attenuated, consistent with the fact that EASY windows already admit strong forecasts from numerical history alone. On the *accepted* set, providing the original (aligned) context reduces CRPS relative to the no-context baseline across most forecasters, and this improvement is reflected by higher win-rates. However, win-rates on EASY tend to remain closer to the equal-performance baseline, indicating smaller margins over Chronos even when context is informative. The *swapped-context* control again degrades performance, increasing CRPS and lowering win-rate compared to the aligned setting, which supports the interpretation that gains arise from instance-level context alignment rather than superficial properties of the text. Finally, on the *rejected* set, adding context typically worsens CRPS and reduces win-rate relative to no-context, as expected given that these windows are explicitly those where the verifier does not benefit from context under the acceptance criterion. Overall, the EASY results corroborate the benchmark construction: accepted windows exhibit measurable context complementarity (albeit with less headroom than HARD), while rejected windows provide a counterfactual control where context is non-informative or misleading.

## F.2. Scaling Laws: Training DoubleCast with Varying Data Fractions

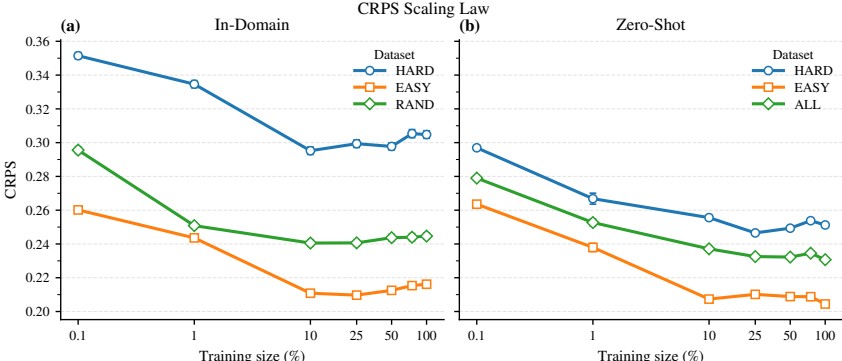

*Figure 24.* Effect of dataset scaling on CRPS performance.

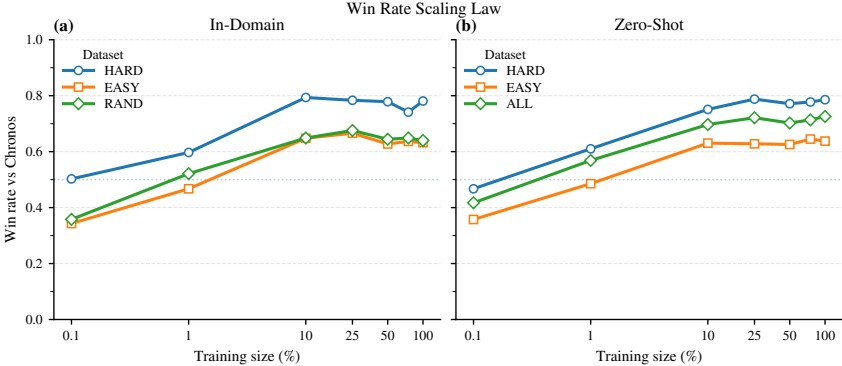

*Figure 25.* Effect of dataset scaling on win rate across models.

Figs. 24–25 report a data-scaling study where DoubleCast is trained with increasing fractions of `indomain-train` (x-axis: training size in %), and evaluated on both in-domain and zero-shot splits, stratified by difficulty. Overall, increasing the training fraction consistently improves performance: mean CRPS decreases as more training windows are used (Fig. 24), and win-rate versus Chronos increases in parallel (Fig. 25). The largest gains occur when moving from very small regimes (0.1%–1%) to moderate regimes (10%–25%), after which improvements tend to taper, indicating diminishing returns at

larger data fractions. Because the training corpus is not forecaster-filtered, this saturation should not be interpreted as a definitive data ceiling; it may instead reflect noise saturation from weakly aligned or uninformative generated contexts. These trends are observed both in-domain and under distribution shift on the zero-shot benchmark.

### F.3. Time-MMD Evaluation

We additionally evaluate DoubleCast on Time-MMD (Liu et al., 2024a), an external multimodal forecasting benchmark with nine domains. Table 7 reports domain-level MAE for Chronos, Informer, zero-shot DoubleCast, and fine-tuned DoubleCast. We omit aggregate MAE because the domain scales differ substantially and the aggregate is dominated by the Security series.

*Table 7.* MAE ($\downarrow$) on Time-MMD by domain. The best value in each row is bolded.

| Domain | Chronos | Informer | DoubleCast-ZS | DoubleCast-FT |
|---|---|---|---|---|
| Agriculture | **3.9850** | 158.2635 | 4.5215 | 4.2144 |
| Climate | 0.5691 | **0.4316** | 0.5596 | 0.4959 |
| Economy | 3,239.3511 | 8,340.6396 | 3,286.1703 | **2,714.7458** |
| Energy | 0.3835 | 0.6271 | 0.4279 | **0.3503** |
| Environment | 23.7829 | **22.2747** | 23.9181 | 22.5789 |
| Public Health | 1.3239 | 1.2432 | 1.2746 | **1.0042** |
| Security | $1.6891 \times 10^9$ | $2.8261 \times 10^9$ | $1.7187 \times 10^9$ | $1.7134 \times 10^9$ |
| SocialGood | 0.8368 | 1.7703 | 0.7889 | **0.7435** |
| Traffic | 23,412.2046 | 237,104.2383 | 25,177.8823 | **19,447.7847** |

Fine-tuning improves over zero-shot DoubleCast on all nine domains by MAE and on eight of nine domains by MSE. It also outperforms Chronos on seven of nine domains by MAE and eight of nine domains by MSE. In contrast, zero-shot DoubleCast beats Chronos on only three of nine domains by MAE, suggesting that Time-MMD requires task adaptation rather than direct transfer from CAF-7M.

At the same time, DoubleCast is not state-of-the-art on Time-MMD: its mean rank is 9.11 by MAE and 10.78 by MSE, while strong numerical forecasting backbones such as iTransformer, PatchTST, and FEDformer rank higher. This supports using Time-MMD as an external adaptation benchmark, but suggests that it is less diagnostic of localized context usage than of forecasting backbone strength.

### F.4. Additional Baselines on CAF-7M

As an additional multimodal reference point, Aurora (Wu et al., 2026) achieves an ALL-split CRPS of 0.589 with context and 0.643 with shuffled context on our test set, corresponding to a 9.2% degradation under context shuffling.

Table 8 reports additional CRPS results for Chronos-2 (Ansari et al., 2025), FlowState (Graf et al., 2025), TimeMixer++ (Wang et al., 2025b), and Multi-Patch Prediction/aLLM4TS (Bian et al., 2024). Chronos-2 is the strongest of these additional baselines across ALL, HARD, and EASY, while FlowState remains close to Chronos and TimeMixer++/Multi-Patch Prediction perform substantially worse on this benchmark.

*Table 8.* Additional baseline CRPS ($\downarrow$) on CAF-7M. The best value in each column is bolded.

| Model | ALL | HARD | EASY |
|---|---|---|---|
| Chronos | 0.278 | 0.304 | 0.247 |
| Chronos-2 | **0.262** | **0.285** | **0.234** |
| FlowState | 0.296 | 0.310 | 0.279 |
| TimeMixer++ | 0.554 | 0.521 | 0.592 |
| LLM4TS (Multi-Patch) | 0.573 | 0.535 | 0.618 |

## F.5. DP-based Context Verification

### F.5.1. VERIFICATION RESULTS

For both `zeroshot-test` (HARD) and `zeroshot-test` (EASY), we visualize the DP-based verification process using (i) *Acceptance funnel* plots (Figs. 26, 28) report how many candidate windows are (a) processed, (b) pass the lightweight semantic judge (`judge_passes` = Q1 & Q2), and (c) are ultimately accepted by DP verification; each stage is annotated with the corresponding percentage of the processed total. (ii) *Per-dataset breakdown* plots (Figs. 27, 29) aggregate outcomes by dataset and report: counts of judge-passed and DP-accepted windows; the judge pass rate (passed divided by total); and the "context helps" rate (accepted divided by passed).

**`zeroshot-test` (HARD).**

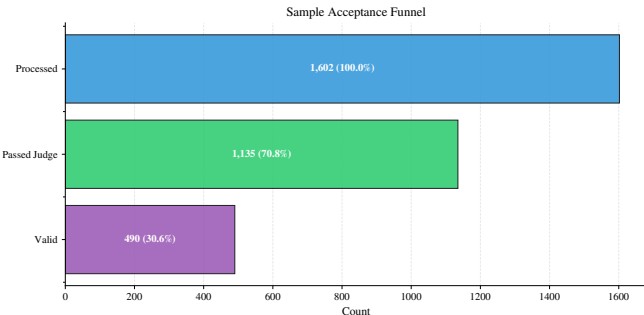

*Figure 26.* Acceptance funnel for DP-based verification on `zeroshot-test` (HARD).

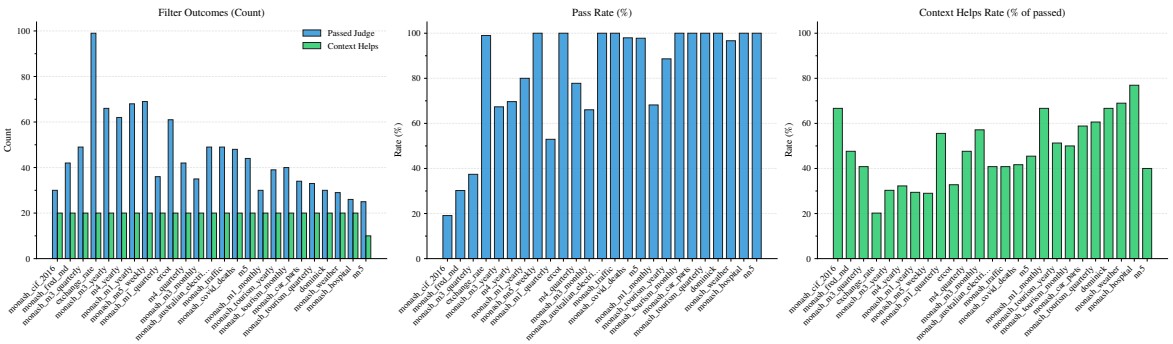

*Figure 27.* Per-dataset breakdown of accepted/rejected windows under DP-based verification on `zeroshot-test` (HARD).

**`zeroshot-test` (EASY).**

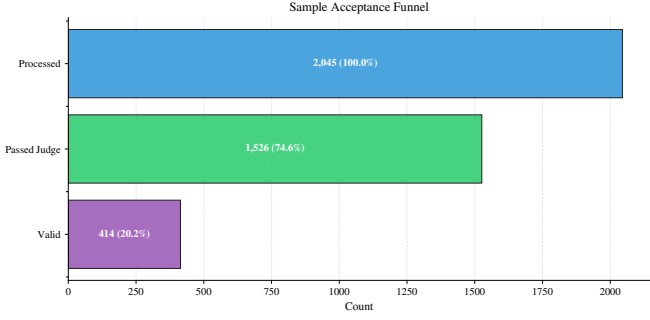

*Figure 28.* Acceptance funnel for DP-based verification on `zeroshot-test` (EASY).

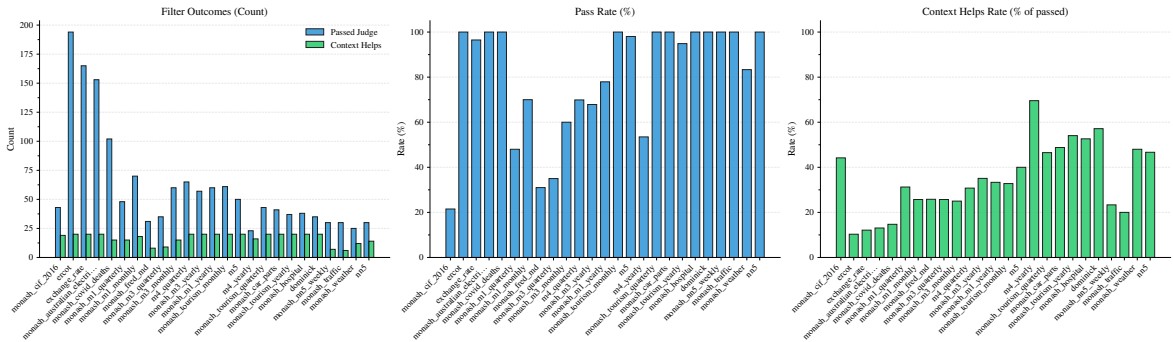

*Figure 29.* Per-dataset breakdown of accepted/rejected windows under DP-based verification on `zeroshot-test` (EASY).

### F.5.2. SUMMARY STATISTICS BEFORE VS. AFTER FILTERING

This subsection tests whether DP-based verification (and associated filtering) induces unintended selection bias toward particular window characteristics. For each evaluation setting (`indomain-test`: ALL/HARD/EASY; `zeroshot-test`: HARD/EASY), we compare the *filtered* split against a *random* baseline of identical size, drawn uniformly from the same candidate pool prior to verification. Each figure overlays normalized histograms (density) of three window-level attributes: the baseline difficulty proxy, history length, and forecast-horizon length. Across both in-domain (Figs. 31–33) and zero-shot (Figs. 34, 35), the filtered distributions closely track their size-matched random counterparts, indicating that verification does not systematically skew the splits toward atypical history lengths or horizons. Deviations in the Chronos MASE panel are expected when conditioning on HARD/EASY, since these variants explicitly stratify windows by the difficulty proxy; importantly, this stratification does not propagate into shifts in history length or forecast length. Together, these checks support that the DP acceptance criterion primarily filters by the predictive usefulness of context rather than by trivial window geometry or sampling artifacts.

### F.6. Impact of the Choice of LLM for Filtering

As described in App. A.1, the CAF-7M testing split was filtered using `GPT-5.2`. To determine how much of an impact on the final filtered dataset this choice of LLM has, we compare the results of this filtering step using `GPT 5.2` versus using 3 alternative LLMs: `Deepseek V3.2`, `Claude Sonnet 4.6`, and `Gemini 2.5 Flash`. Due to increased cost of the DP verification step, this experiment was done on a subset of 200 windows from the HARD set.

As can be seen in Fig. 30a, all 4 models have very high agreements about whether the windows pass the semantic judge filtering step. The lowest agreement with `GPT 5.2` are `Claude Sonnet 4.6` and `Gemini 2.5 Flash`, both with an agreement of 89.2%. This high agreement points that this step of the filtering process is mostly unaffected by the choice of LLM, as long as the LLM chosen is strong enough at the task.

However, Fig. 30b shows that the opposite is true for the DP verification step, with an agreement with `GPT 5.2` that goes from 43.5% to only 54.0%. This indicates that, unlike for the semantic judge step, the choice of LLM for the DP verification is crucial, since the set of windows which passes this step would be completely different if another LLM was to be used instead of `GPT 5.2`. Nevertheless, this experiment shows that among those 4 models, `GPT 5.2` is indeed the LLM that is best suited for this task. Indeed, in idealized circumstances, this verification step would use the best possible context-aided

*Table 9.* How often the forecasts produced using Direct Prompt with different models has the lowest CRPS values amongst their peers. These results are using 200 windows from a partially filtered version of CAF-7M HARD test set, after the semantic filtering step but prior to the DP verification step.

| Model | Win rate without context (%) | Win rate with context (%) |
|---|---|---|
| GPT 5.2 | 23.5 | **46.0** |
| Deepseek V3.2 | 27.5 | 18.5 |
| Claude Sonnet 4.6 | 22.5 | 17.5 |
| Gemini 2.5 Flash | 25.5 | 18.0 |

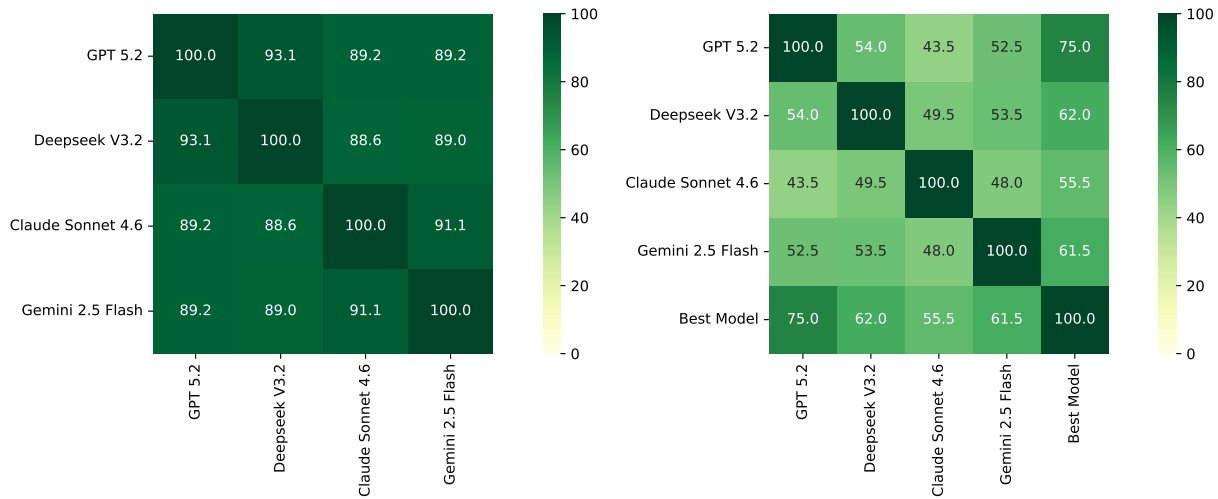

*(a)* Agreement rates for the semantic judge.

*(b)* Agreement rates for the DP verification.

*Figure 30.* Comparison of how often various LLMs agreed about whether a window was considered valid by the semantic judge or by the DP verification steps. *Best Model* represents using the smallest CRPS value from all 4 models instead of using a single model.

forecasting method which can be approximated by the *Best Model* version, which uses the most accurate forecast (lowest CRPS) from the forecasts from the 4 models. Since `GPT 5.2` has the highest agreement (75.0%) with *Best Model*, it is thus the most appropriate model for the DP verification. To understand why `GPT 5.2` agrees more often with *Best Model*, Tab. 9 shows how often its forecasts are the most accurate: while `GPT 5.2` is not superior to the other LLMs when forecasting without context, it is definitely superior on context-aided forecasting.

`indomain-test.`

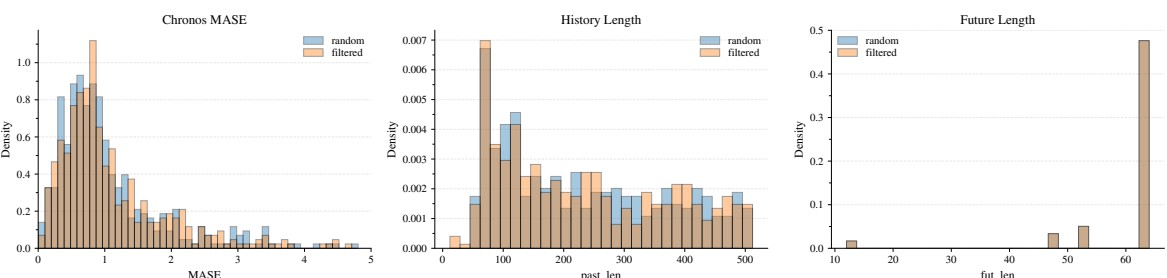

*Figure 31.* `indomain-test` (ALL): summary statistics before vs. after filtering.

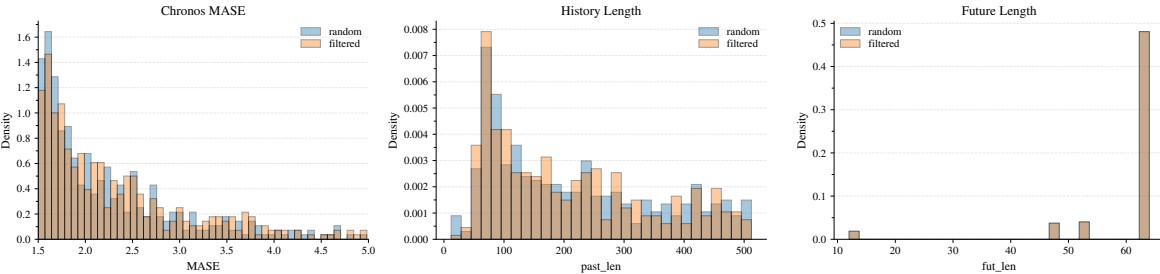

*Figure 32.* `indomain-test` (HARD): summary statistics before vs. after filtering.

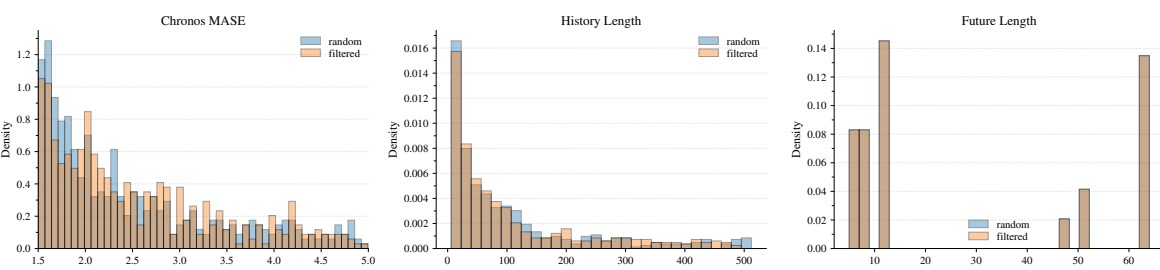

*Figure 33.* `indomain-test` (EASY): summary statistics before vs. after filtering.

**zeroshot-test.**

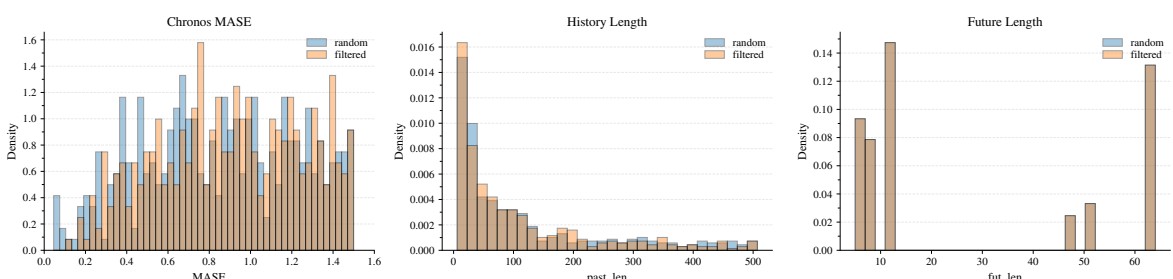

*Figure 34.* `zeroshot-test` (HARD): summary statistics before vs. after filtering.

*Figure 35.* `zeroshot-test` (EASY): summary statistics before vs. after filtering.

## F.7. Qualitative Forecast Examples: When Context Helps vs. Hurts

### F.7.1. ZEROSHOT-TEST (HARD): CONTEXT HELPS

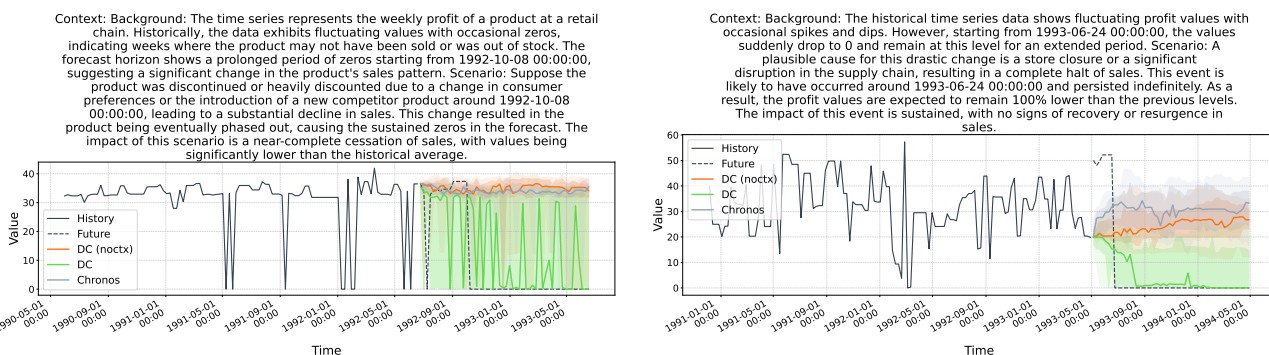

*Figure 36.* Examples of windows in `zeroshot-test` (HARD) where the context was found to be helpful when forecasting using DoubleCast.

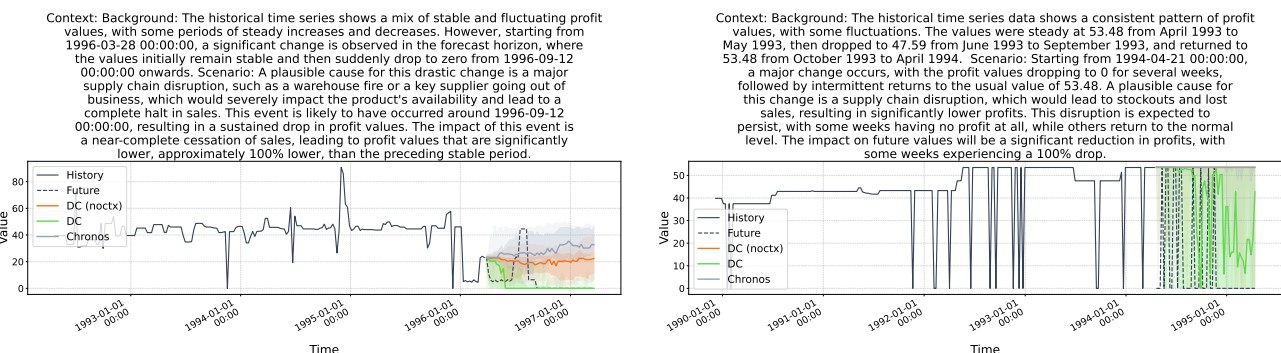

*Figure 37.* Examples of windows in `zeroshot-test` (HARD) where the context was found to be helpful when forecasting using DoubleCast.

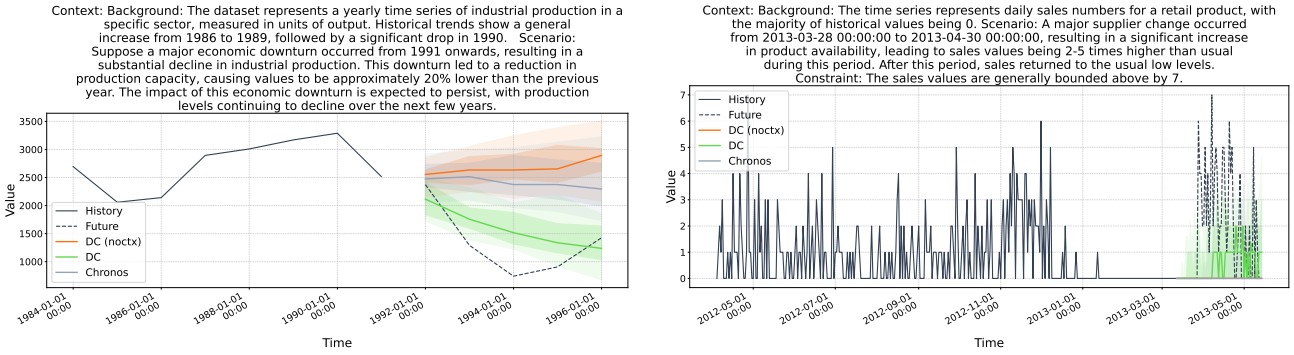

*Figure 38.* Examples of windows in `zeroshot-test` (HARD) where the context was found to be helpful when forecasting using DoubleCast.

*Figure 39.* Examples of windows in `zeroshot-test` (HARD) where the context was found to be helpful when forecasting using DoubleCast.

*Figure 40.* Examples of windows in `zeroshot-test` (HARD) where the context was found to be helpful when forecasting using DoubleCast.

### F.7.2. `ZEROSHOT-TEST` (HARD): CONTEXT HURTS

*Figure 41.* Examples of windows in `zeroshot-test` (HARD) where the context was found to not be helpful when forecasting using DoubleCast.

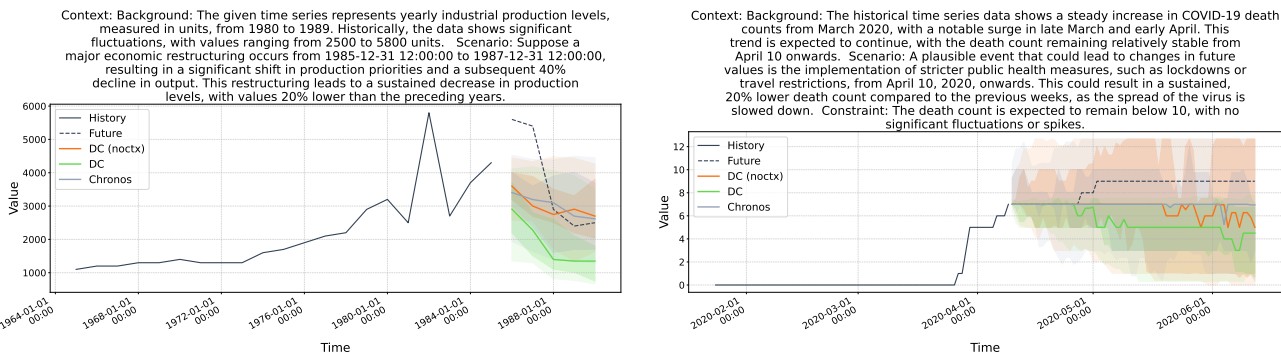

*Figure 42.* Examples of windows in `zeroshot-test` (HARD) where the context was found to not be helpful when forecasting using DoubleCast.

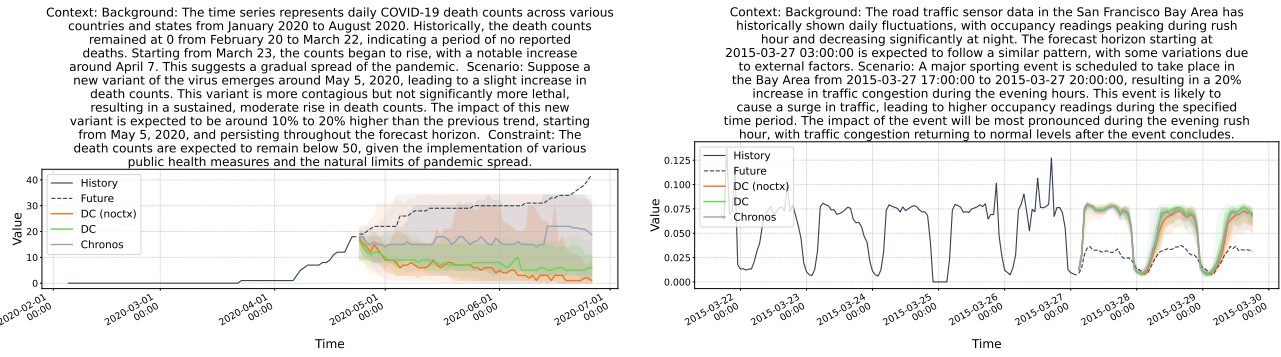

*Figure 43.* Examples of windows in `zeroshot-test` (HARD) where the context was found to not be helpful when forecasting using DoubleCast.

*Figure 44.* Examples of windows in `zeroshot-test` (HARD) where the context was found to not be helpful when forecasting using DoubleCast.

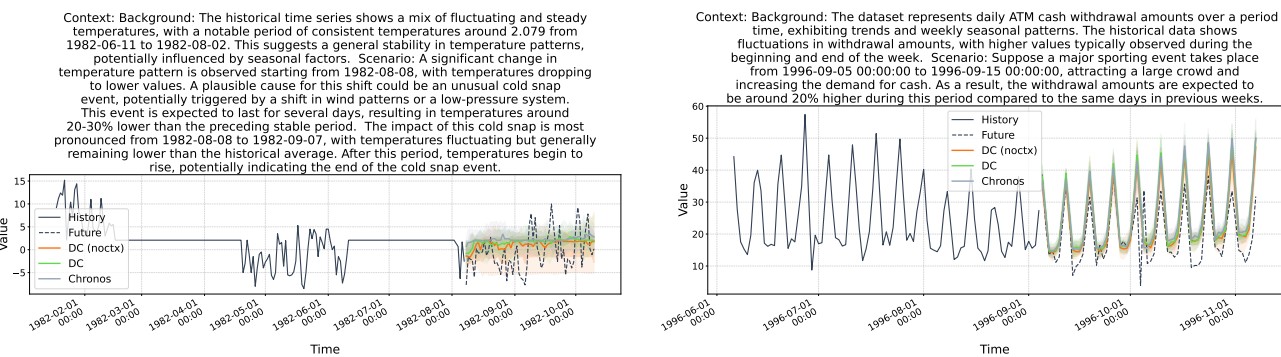

*Figure 45.* Examples of windows in `zeroshot-test` (HARD) where the context was found to not be helpful when forecasting using DoubleCast.

### F.7.3. `ZEROSHOT-TEST` (EASY): CONTEXT HELPS

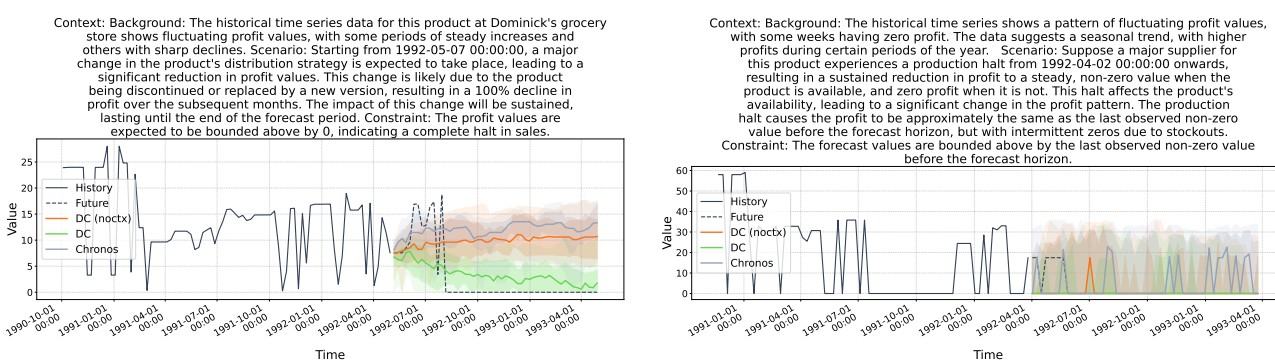

*Figure 46.* Examples of windows in `zeroshot-test` (EASY) where the context was found to be helpful when forecasting using DoubleCast.

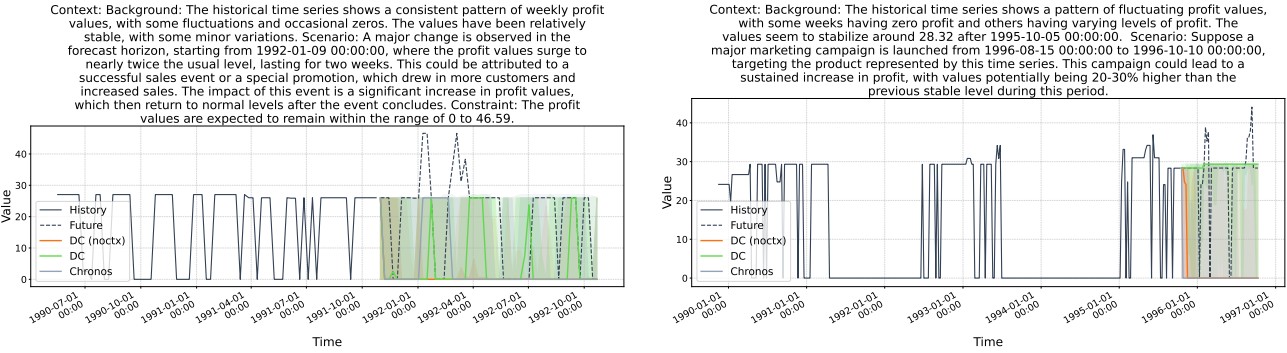

*Figure 47.* Examples of windows in `zeroshot-test` (EASY) where the context was found to be helpful when forecasting using DoubleCast.

*Figure 48.* Examples of windows in `zeroshot-test` (EASY) where the context was found to be helpful when forecasting using DoubleCast.

*Figure 49.* Examples of windows in `zeroshot-test` (EASY) where the context was found to be helpful when forecasting using DoubleCast.

*Figure 50.* Examples of windows in `zeroshot-test` (EASY) where the context was found to be helpful when forecasting using DoubleCast.

## F.7.4. `ZEROSHOT-TEST` (EASY): CONTEXT HURTS

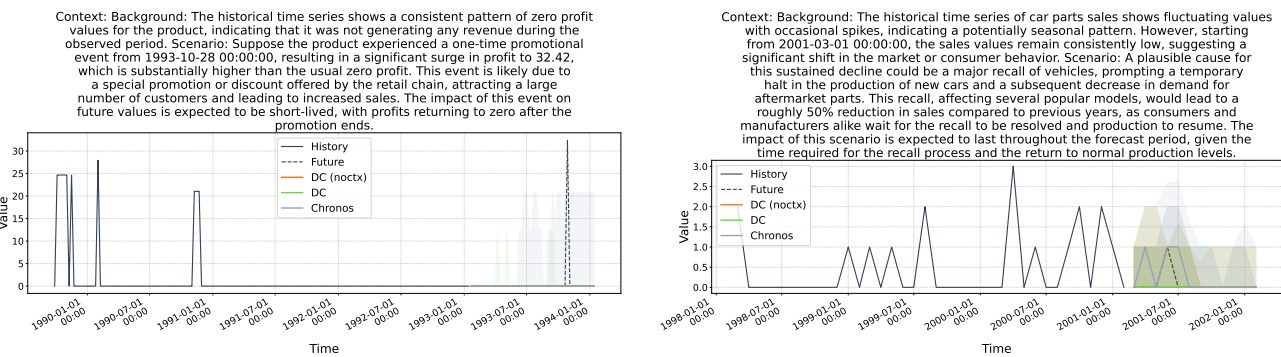

*Figure 51.* Examples of windows in `zeroshot-test` (EASY) where the context was found to not be helpful when forecasting using DoubleCast.

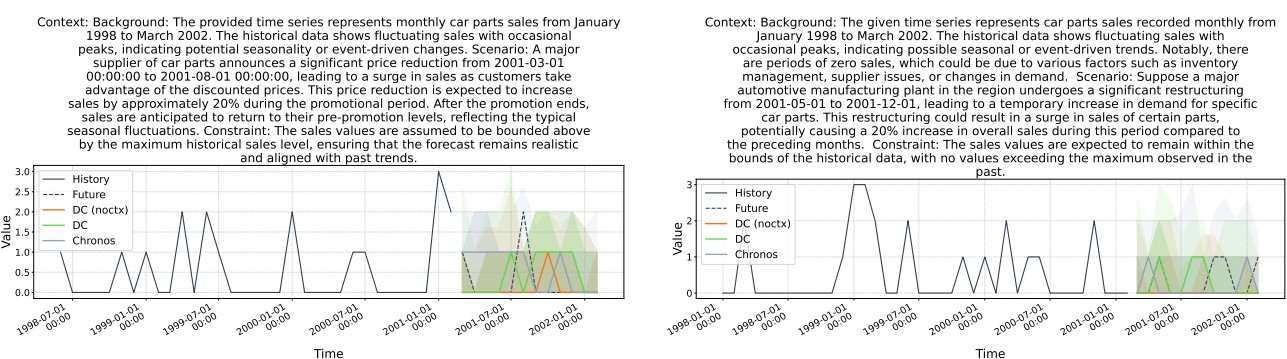

*Figure 52.* Examples of windows in `zeroshot-test` (EASY) where the context was found to not be helpful when forecasting using DoubleCast.

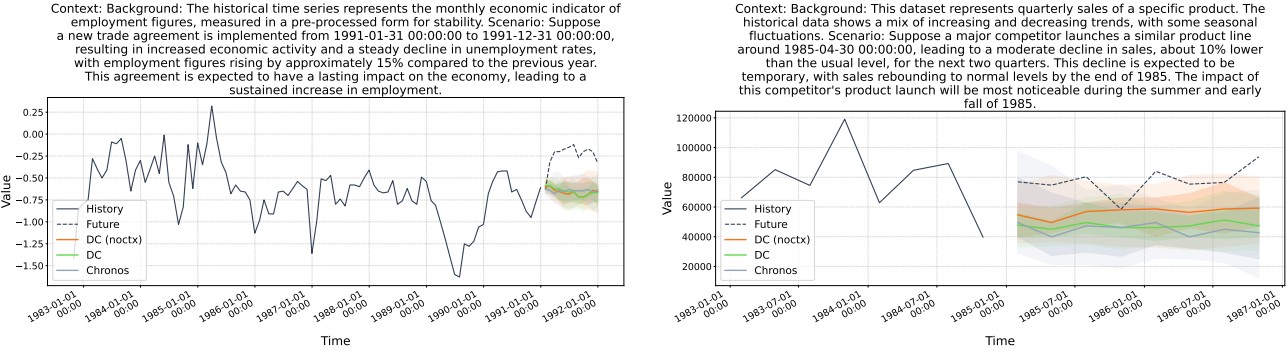

*Figure 53.* Examples of windows in `zeroshot-test` (EASY) where the context was found to not be helpful when forecasting using DoubleCast.

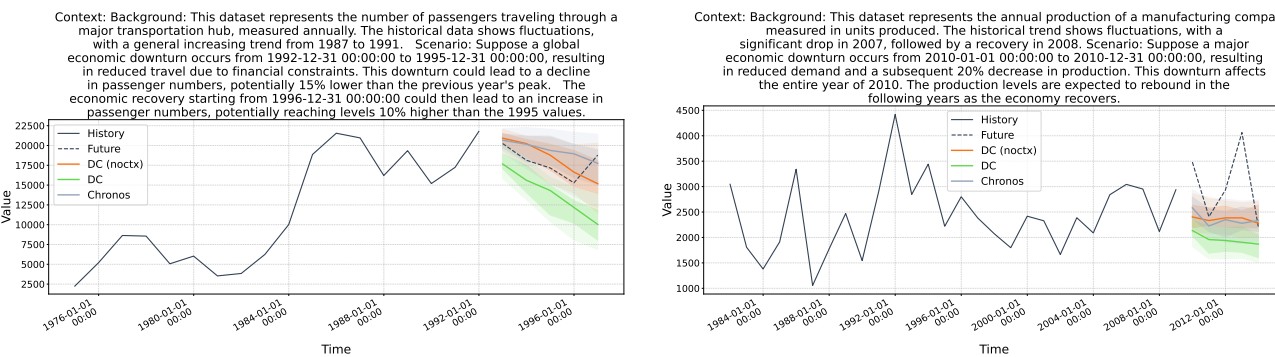

*Figure 54.* Examples of windows in `zeroshot-test` (EASY) where the context was found to not be helpful when forecasting using DoubleCast.

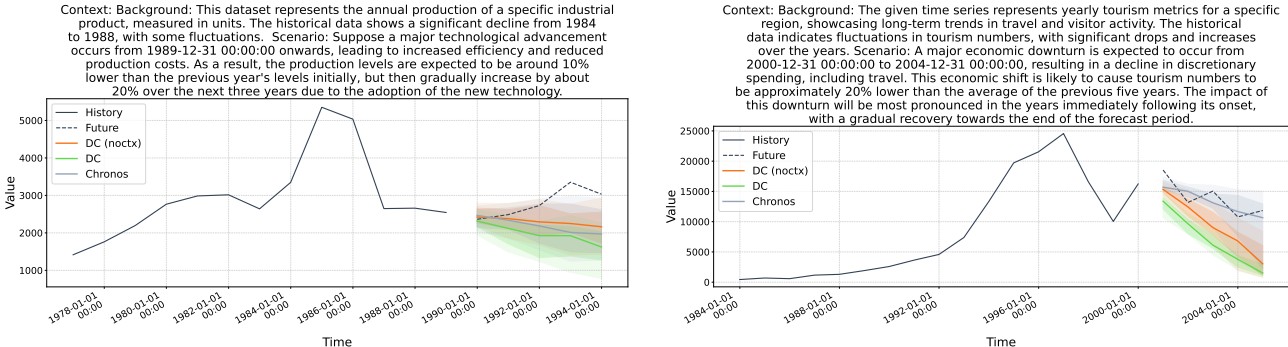

*Figure 55.* Examples of windows in `zeroshot-test` (EASY) where the context was found to not be helpful when forecasting using DoubleCast.

