# OpenReview forum: "Overcoming the Modality Gap in Context-Aided Forecasting"
_ICML.cc/2026/Conference — ICML 2026 regular_

### Official Review · Reviewer_9U3r · 2026-02-18

**Soundness:** 3
**Presentation:** 3
**Significance:** 3
**Originality:** 3
**Overall Recommendation:** 4
**Confidence:** 4

**Summary:**

This paper investigates a frustrating puzzle in context-aided forecasting (CAF): why do multimodal models that fuse text with time series often fail to beat unimodal baselines? The authors' hypothesis is refreshingly simple—it's not the architectures, it's the data. They test this with a two-phase semi-synthetic pipeline: first, Llama-3-70B generates scenario-style textual contexts conditioned on both historical and future values of time series windows; then, GPT-5.2 via Direct Prompt filters contexts by keeping only those that demonstrably improve CRPS. The result is CAF-7M, a 7.4M-window corpus from 65 datasets across 11 domains, including 904 rigorously verified test instances (split into HARD and EASY based on Chronos difficulty). They validate the dataset with DoubleCast, which adds DualT5 cross-attention layers over Qwen3-14B text embeddings to the Chronos foundation model. The experiments show genuine context utilization (strong degradation when context is removed/swapped), real-world transfer (CGTSF), and preserved numerical ability (GIFT-Eval).

**Compliance With Llm Reviewing Policy:**

Affirmed.

**Final Justification:**

Overall, this paper made a non-negligible contribution to the community. Thus, I maintain my positive score.

**Key Questions For Authors:**

None.

**Limitations:**

The authors provide an excellent limitations section (Section 5) covering verifier bias, unfiltered training data, context diversity, and DoubleCast's temporal localization failures. This is above the ICML average. Additional points: (1) the pipeline's computational cost (Llama-3-70B + GPT-5.2) may limit accessibility; (2) the methodology needs future values for generation, so it can't be directly applied in real-time deployment deserves mention.

**Strengths And Weaknesses:**

### Strengths

1. The research question is both important and well-framed. The failure of multimodal CAF models to beat unimodal baselines has been a persistent frustration, and this paper is the first I've seen to systematically investigate why. The hypothesis that data quality is the bottleneck rather than architecture is elegant and, based on the evidence, largely convincing. If this holds up, it redirects a lot of wasted effort on architecture design toward data curation—a genuinely useful reorientation for the field.
2. The generate-then-verify methodology is the paper's strongest contribution. Formally defining "useful context" via proper scoring rules (Eq. 2) and then filtering based on actual forecasting improvement is clean and principled. What really sold me was the swapped context experiment (Figure 3): showing that mismatched contexts don't help and actually hurt is strong evidence that gains come from genuine complementary information, not just from having any text present. The HARD/EASY split based on Chronos MASE is another nice design choice—without it, the test set would be dominated by easy cases where context barely matters.
3. The scale is impressive and the transfer results matter. 7.4M windows from 65 datasets across 11 domains is far larger than anything else available (Table 1). But what really matters is whether semi-synthetic pretraining generalizes to real data—and Table 3's CGTSF results show it does. DoubleCast finetuned on CGTSF outperforms both Chronos and ChatTime, which is the key practical validation.
4. The evidence for context utilization is clean. Table 2 shows DoubleCast's CRPS degrades by 18.3% without context and 26.5% with swapped context, with corresponding Win Rate drops. This is much stronger than TimeLLM's pattern, where removing/swapping context sometimes helps. The GIFT-Eval results (Figure 4) confirming no sacrifice in numerical forecasting ability complete the picture.
5. The limitations section (Section 5) is unusually honest and thorough—verifier bias, unfiltered training data, context diversity, temporal localization failures are all acknowledged. This kind of transparency is rare and I appreciate it.

### Weaknesses

1. The verification circularity is a genuine concern. GPT-5.2 via Direct Prompt decides what counts as "useful" context, so the test set is inherently biased toward context patterns this specific model can exploit. Other architectures might benefit from different kinds of context that GPT-5.2 misses. The authors show in Figure 3 that accepted/rejected labels correlate across forecasters, which helps, but a more systematic analysis—comparing verification decisions from 2-3 different verifier architectures and reporting overlap statistics—would go a long way toward resolving this.
2. The training data is unfiltered while the test data is curated, and this asymmetry is underexplored. The 7.4M training contexts presumably contain plenty of noise (misleading, vacuous, or wrong contexts). DoubleCast learns effectively anyway, which suggests robustness, but the paper never asks how much better results could be with cleaner training data. Even a small experiment—filtering 10% of training data and comparing—would be informative. I also wonder whether the diminishing returns in the scaling analysis (Appendix F.2) reflect noise saturation rather than a genuine ceiling.
3. The unimodal baselines are limited to Chronos and AutoARIMA. The paper argues that context provides value beyond numerical history, but to make this case convincingly, you need to compare against the strongest possible numerical-only methods. Recent approaches like Multi-Patch Prediction (Bian et al., ICML 2024) and TimeMixer++ (Wang et al., ICLR 2025) learn richer temporal representations that might already capture some of the signal attributed to context (e.g., regime changes detectable from numerical patterns alone). Without these comparisons, the marginal value of text remains somewhat uncertain.


### References

- Wang, S., Li, J., Shi, X., Ye, Z., Mo, B., Lin, W., Ju, S., Chu, Z., & Jin, M. (2025). TimeMixer++: A General Time Series Pattern Machine for Universal Predictive Analysis. In *International Conference on Learning Representations (ICLR 2025)*.
- Bian, Y., Ju, X., Li, J., Xu, Z., Cheng, D., & Xu, Q. (2024). Multi-patch prediction: Adapting language models for time series representation learning. In *Forty-first International Conference on Machine Learning (ICML 2024)*.

---

> ### Author Rebuttal · Authors · 2026-03-31
>
> We thank Reviewer 9U3r for the thorough review. We address each point below.
>
> ---
>
> ### W1: Verification circularity.
>
> We agree that this is a valid concern. We will make sure to **note in the revised version of the paper and on the public release of the dataset** that GPT-5.2 (and probably also other models from its family) has an innate advantage on our test set due to having been used in the filtering process. So any benchmark comparing GPT-5.2 with other models using CAF-7M should mention it as a disclaimer.
>
> We did run your suggested experiments on the **LLM-as-a-Judge** portion of the filtering process. We obtained **strong agreement between GPT-5.2 decisions and the 3 other models we tried**: Deepseek V3.2 at 92.3%, Claude Sonnet 4.6 at 88.3%, and Gemini 2.5 Flash at 88.3%. Sadly, comparing the forecasting quality portion of the filtering process turned out to be prohibitive, especially for Claude Sonnet.
>
>
> ### W2: Unfiltered training vs. curated test is underexplored.
>
> In an earlier in-domain validation experiment, we observed a clear **quantity-versus-quality trade-off**. At ~40k steps, the approximate in-domain validation cross-entropy losses were as follows (lower is better; gap is relative to the matched unfiltered run):
>
> | Budget | filtered @ 40k | unfiltered @ 40k | gap (%) |
> | --- | ---: | ---: | ---: |
> | 50k | ~2.89 | ~2.95 | +2.0% |
> | 25k | ~3.10 | ~3.24 | +4.3% |
> | 10k | ~3.63 | ~3.75 | +3.2% |
> | 5k | ~4.11 | ~4.20 | +2.1% |
>
> This suggests that filtered data is more sample-efficient at matched budgets, while **larger unfiltered sets can  recover and even exceed the gap** (e.g., `unfiltered_50k` outperforms `filtered_25k`). However, this filtering used a much cheaper LLM than the GPT-5.2-based filtering used for the test set, so we view this result as suggestive rather than definitive. This trade-off is one reason we chose to keep the training set unfiltered. We also note that the scaling-curve saturation observed in our experiments may indeed reflect noise saturation rather than a true data ceiling, and **we will discuss this interpretation in the revision**.
>
> ### W3: Unimodal baselines are limited.
>
> Thank your for your suggestion. Based on the GIFT-Eval leaderboard, we have **added two strong zero-shot unimodal baselines: Chronos-2 and Flowstate**. We find that both of them perform similar or worse than Chronos (full results in Table 2 of the paper).
>
> | Model | Zero-Shot/ALL | Zero-Shot/HARD | Zero-Shot/EASY |
> |---|---:|---:|---:|
> | Chronos-2 | 0.26182 ± 0.00270 | 0.28525 ± 0.00236 | 0.23409 ± 0.00520 |
> | FlowState | 0.29552 ± 0.00215 | 0.30986 ± 0.00125 | 0.27856 ± 0.00445 |
>
>
> We agree that other numerical-only baselines such as Multi-Patch Prediction and TimeMixer++ would strengthen the case for multimodal context. As these models are not zero-shot foundation models, they require unimodal training. Before the camera-ready version, **we will consider how to fairly train and evaluate such models**.
>
>
> ---
>
> ### Additional limitations noted by reviewer:
>
> ### Pipeline computational cost / accessibility.
>
> We acknowledge that reproducing the full pipeline may not be accessible to all research groups. **We will release the dataset, generation prompts, and verification code** to maximize reproducibility without requiring re-running the full pipeline.
>
> ### Needs future values during generation, so it cannot be directly applied in real-time deployment.
>
> We agree that using future values limits the direct deployability of the generation pipeline itself. However, our goal is to train and evaluate models that can use arbitrary forward-looking free-form context available at forecast time, such as expert expectations or known upcoming events. The future-conditioned generation is therefore an **offline data-construction tool, not a deployment-time requirement of the model**.

---

> > ### Author Rebuttal · Reviewer_9U3r · 2026-04-02
> >
> > Thank you for the thorough rebuttal. The cross-verifier experiment with three additional models (Deepseek V3.2, Claude Sonnet 4.6, Gemini 2.5 Flash) showing 88–92% agreement with GPT-5.2, the quantity-vs-quality training data analysis, and the additional unimodal baselines (Chronos-2, FlowState) collectively strengthen the paper. My concern about verification circularity is partially but not fully resolved, as the forecasting-quality portion of the filtering remains tested with only one verifier; the commitment to evaluate Multi-Patch Prediction and TimeMixer++ for the camera-ready version is noted. I maintain my score.

---

> > > ### Author Response · Authors · 2026-04-07
> > >
> > > We have ran additional experiments using the TimeMixer++ and Multi-Patch Prediction models.
> > > For these results, we trained TimeMixer++ on our CAF-7M training set, using out in-domain validation set to determine when to stop early.
> > > For Multi-Patch Prediction, we used the official aLLM4TS and we trained it on LLM4TS_sft_zero.
> > >
> > > Here are the CRPS results for our test set (note that these models are not probabilistic ones, so they are somewhat disadvantaged by our use of CRPS as the metric):
> > >
> > > | Test set | TimeMixer++ CRPS | LLM4TS / Multi-patch prediction CRPS |
> > > |---|---:|---:|
> > > | CAF-7M test set (ALL) | 0.553936 | 0.572660 |
> > > | CAF-7M test set (HARD) | 0.521422 | 0.534686 |
> > > | CAF-7M test set (EASY) | 0.592487 | 0.617605 |
> > >
> > > ---
> > >
> > > Regarding the issue of verification circularity, we conducted an additional experiment comparing the filtering decisions based on the forecast CRPS from GPT 5.2, Deepseek V3.2, Claude Sonnet 4.6, and Gemini 2.5 Flash. This evaluation was performed on a 200-sample subset of our unfiltered test set, assessing forecasts both independently and with context.
> > >
> > > Because baseline inter-model agreement is moderate (ranging from 43.5% to 54.0%), it is clear that model selection significantly impacts the filtering pipeline. Against this variance, GPT 5.2 stands out by demonstrating a high 75% agreement with the best-case baseline (i.e., selecting the lowest CRPS achieved by any of the four models for a given sample).
> > >
> > > This strong alignment is explained by GPT 5.2 being empirically our most capable context-aided forecaster, yielding the best forecasts for 46.0% of the samples when context is provided. In contrast, when context is withheld, performance is evenly distributed across the ensemble, with GPT 5.2 leading in only 23.5% of cases. Consequently, our choice to utilize GPT 5.2 in the filtering procedure does not introduce an arbitrary bias; rather, it is an empirically justified design choice that establishes the most rigorous possible benchmark for context-driven forecasting capabilities.

---

### Official Review · Reviewer_MBbq · 2026-02-28

**Soundness:** 2
**Presentation:** 2
**Significance:** 2
**Originality:** 2
**Overall Recommendation:** 3
**Confidence:** 5

**Summary:**

This paper introduces a context generation methodology that enables researchers to convert any numerical time series dataset into a context-aided forecasting dataset, with provably useful context. The constructed dataset CAF-7M contains 7,433,239 context-augmented time series windows for training and 904 for testing. Experimental results on DoubleCast, a new multi-modal model architecture, demonstrate the usefulness of CAF.

**Compliance With Llm Reviewing Policy:**

Affirmed.

**Final Justification:**

Thanks for the response. However, I remain unconvinced regarding the direct exclusion of existing multimodal TSFMs from both the verification stage and experiment stage. Without addressing or comparing against the multimodal paradigm, it is difficult to assess the actual efficacy and practical applicability of the proposed method within the broader research landscape. So I keep my assessment as a weak reject.

**Key Questions For Authors:**

yes

**Limitations:**

See Weakness.

**Strengths And Weaknesses:**

**Strengths**

1. Constructing a context-augmented time series dataset is beneficial for the community.
2. The proposed DoubleCast is promising and effective.
3. Comprehensive evaluations demonstrate the generalizability of CAF to real-world context-aided forecasting datasets.

**Weakness**

1. In the generation phase, the authors prompt the LLM to generate contexts that are both descriptive of temporal dynamics and verifiably complementary to numerical histories. As this is an LLM-powered method, I am curious how the authors addressed potential LLM hallucinations. Relying solely on the LLM-based verification method may raise questions about reliability.
2. During the verification stage, the authors use the Direct Prompt forecaster to check compliance. Could the authors clarify why they chose these methods over current multimodal time series forecasting models?
3. In Table 2, DoubleCast performs worse than Direct Prompt (DP) with GPT-5.2, likely because the test set was pre-filtered by DP. However, DP is missing as a baseline in Table 3, which is intended to evaluate generalizability across real-world benchmarks.

---

> ### Author Rebuttal · Authors · 2026-03-31
>
> We thank Reviewer MBbq for the careful reading and constructive feedback. We address each point below.
>
> ---
>
> ### W1: Potential LLM hallucinations and reliability of LLM-based verification.
>
> We emphasize that **the main verification step is quantitative rather than purely judge-based**: a window is retained when the Direct Prompt verifier achieves lower CRPS with the generated context than without it. Before this forecast step, we also apply a lightweight **semantic validity check** to filter clearly invalid contexts. In addition, we conducted a **human evaluation** with 11 annotators: majority-vote judgments were positive for 80% of examples on whether the context background was clear and 82% on whether the context scenario was plausible. Hallucinations may still exist in the generated text, but for our benchmark the primary criterion is measured by forecasting improvement rather than surface realism alone.
>
> ### W2: Why use Direct Prompt rather than current multimodal time-series forecasting models for verification?
>
> Direct Prompt [1] works out of the box in a zero-shot manner, whereas many existing multimodal time-series models are not drop-in verifiers [2]. This makes DP **the most straightforward zero-shot choice for verification**.
>
> ### W3: DP is in Table 2 but missing in Table 3.
>
> We acknowledge this gap. Table 3 covers 3 datasets with 4 history/prediction settings each, i.e., 12 evaluation settings in total. Running Direct Prompt across all of these settings would incur **substantial monetary cost**. In light of Table 2, we would indeed expect DP GPT-5.2 to outperform DoubleCast, but the primary goal of Table 3 is to show how much of the CGTSF contexts DoubleCast leverages to improve its forecasts over Chronos forecasts.
>
> [1] Williams, Andrew Robert, et al. "Context is Key: A Benchmark for Forecasting with Essential Textual Information." arXiv:2410.18959. ICML 2025.
>
> [2] Zhang, Xiyuan, et al. "When Does Multimodality Lead to Better Time Series Forecasting?" arXiv preprint arXiv:2506.21611 (2025).

---

> > ### Author Rebuttal · Reviewer_MBbq · 2026-04-02
> >
> > Thanks for the rebuttal. I also remain unconvinced by verification relying solely on a Direct Prompting. Given the emergence of zero-shot multimodal time-series foundation models, such as Aurora, I am concerning that the inherent differences in language understanding capabilities between LLM-based forecasters and native Multimodal TSFMs might results in biased evaluation. I would welcome a deeper discussion on the potential performance gap and the underlying reasons.

---

> > > ### Author Response · Authors · 2026-04-07
> > >
> > > Thank you for your comments. There is an inherent difference between multimodal TSFMs and LLM-based methods such as Direct Prompt. Prior work [1] that extensively evaluates multimodal TSFMs for contextual forecasting suggests that these models are effective mainly when they are trained in-distribution and with context similar to that seen during training. In contrast, LLMs can generally be used off the shelf across a much broader range of contexts. This is the main reason we use Direct Prompt as the verifier rather than a multimodal TSFM: a TSFM may itself be biased in how it uses context, making it difficult to determine whether it truly understands the context, relies on shortcuts, or simply overfits. Although LLM-based verification is still influenced by the model’s underlying forecasting ability, we believe this bias is less severe than in the TSFM case.
> > >
> > >
> > > [1] Zhang, Xiyuan, et al. "When Does Multimodality Lead to Better Time Series Forecasting?" arXiv preprint arXiv:2506.21611 (2025).

---

### Official Review · Reviewer_28cc · 2026-03-06

**Soundness:** 2
**Presentation:** 3
**Significance:** 2
**Originality:** 2
**Overall Recommendation:** 3
**Confidence:** 4

**Summary:**

This work attempts to address a critical question: how to construct datasets where textual context truly helps time series forecasting. The paper proposes a semi-synthetic pipeline that generates scenario-style textual context using a large language model and then verifies its usefulness by keeping only the samples where the context improves forecasting accuracy.

**Compliance With Llm Reviewing Policy:**

Affirmed.

**Final Justification:**

Thank you for your comments. Our paper differs from prior work in two key ways. First, we systematically evaluate whether DoubleCast truly leverages contextual information through both no-context and shuffled-context experiments. Second, our verification step goes beyond a simple LLM-as-a-Judge setup, where the model merely outputs a YES/NO decision on whether the context is complementary. Instead, we assess this quantitatively by measuring the change in CRPS for a GPT-5.2-based Direct Prompt forecaster. To our knowledge, this approach has not yet been used for a time series forecasting usecase.

**Key Questions For Authors:**

1. Since the LLM has access to the future time series during the data generation process, how do you ensure that it does not leak future information in the generated context?

2. Does the fact that the forecast improves when provided with the generated context strictly demonstrate that the context itself is truly useful? It is possible that the model is simply learning shortcuts, such as mapping words like “spike” to an increase and “drop” to a decrease. Ideally, this should be validated across multiple models rather than relying only on a single state-of-the-art model.

**Limitations:**

See weakness and questions.

**Strengths And Weaknesses:**

Strengths:
1. The paper highlights an important research question in context-aided forecasting, namely the modality gap, where textual context cannot be guaranteed to provide useful information for forecasting.
2. The paper provides a relatively thorough empirical evaluation of the proposed dataset and analyzes whether models actually use the contextual information.

Weaknesses:
1. The first two contributions (the data synthesis pipeline and the CAF-7M dataset) appear to have limited novelty. Aurora [1] already uses a similar pipeline to generate contextual information for time series forecasting in multimodal foundation model pretraining. The human verification step mainly demonstrates additional effort rather than conceptual novelty.
2. The proposed DoubleCast architecture also shows limited novelty compared with Aurora.
3. The experimental evaluation is incomplete. In particular, results on the Time-MMD dataset [2] are missing.

[1] Aurora: Towards Universal Generative Multimodal Time Series Forecasting
[2] Time-MMD: Multi-Domain Multimodal Dataset for Time Series Analysis

---

> ### Author Rebuttal · Authors · 2026-03-31
>
> We thank Reviewer 28cc for the thoughtful review. We address each point below.
>
> ---
>
> ### W1: Limited novelty of the data synthesis pipeline and CAF-7M relative to Aurora.
>
> We appreciate the comparison and would like to clarify the distinction. Aurora uses LLM-generated context as a pretraining ingredient, whereas our main contribution is the explicit *generate-then-verify* framework, which verifies generated contexts by measuring their impact on forecasting accuracy (CRPS with vs. without context). **This quantitative step is absent in Aurora's pipeline**.
>
> Also, the paper does not clearly report how many generated contexts were used, and **the corpus itself was unreleased at the time of submission**, preventing direct comparison. According to Aurora's Github, the dataset is actually part of another research project: [1], which was only published on arXiv in February 2026. Nevertheless, the differences and similarities between both pipelines are **discussed in the revised version of our paper**.
>
> To provide additional empirical context, **we evaluated Aurora's trained model on our full test set** ("ALL", Table 2). Aurora achieves a CRPS of 0.589 with context, vs. 0.643 with shuffled context, meaning CRPS worsens by about 9.2% under shuffled context. We report this as a reference point only since Aurora was trained on a different training set.
>
> [1] Chen, Peng, et al. "Empowering Time Series Analysis with Large-Scale Multimodal Pretraining" arXiv preprint arXiv:2602.05646 (2026).
>
> ### W2: Limited novelty of DoubleCast relative to Aurora.
>
> DoubleCast is **intentionally simple and not our core novelty**. It serves as a lightweight multimodal baseline built on top of existing unimodal backbones to isolate the contribution of the CAF dataset. Unlike Aurora, which only reports multimodal results and separate unimodal results under modality absence, we report matched same-instance controls with correct context, no context and swapped context to directly test whether instance-aligned context itself provides measurable forecasting value. **We will therefore reframe DoubleCast as a practical, extensible baseline** rather than as a definitive comparison against the now-cited Aurora to avoid overclaiming.
>
> ### W3: Experimental evaluation is incomplete; Time-MMD is missing.
>
> We agree that Time-MMD is a useful additional benchmark; **we run DoubleCast on it** both zero-shot and fine-tuned. Time-MMD context **does not yield consistent zero-shot gains for a simple multimodal extension**: DoubleCast-ZS beats Chronos on only 3/9 domains in MAE and 4/9 in MSE. In contrast, DoubleCast-FT improves over DoubleCast-ZS on 9/9 domains in MAE and 8/9 in MSE, and over Chronos on 7/9 domains in MAE and 8/9 in MSE.
>
> MAE on Time-MMD (lower is better; bold = best):
>
> | Domain/summary|Chronos|DoubleCast-ZS|DoubleCast-FT |
> | ----| ---: | ----: | ----: |
> | Agriculture|**3.99** |4.52 |4.21 |
> | Climate|0.57 |0.56 |**0.50** |
> | Economy|3,239 |3,286 |**2,715** |
> | Energy|0.38 |0.43 |**0.35** |
> | Environment|23.78 |23.92 |**22.58** |
> | PublicHealth|1.32 |1.27 |**1.00** |
> | Security| **1.69e9** |1.72e9 |1.71e9 |
> | SocialGood|0.84 |0.79 |**0.74** |
> | Traffic|23,412 |25,178 |**19,448** |
> | **Wins vs.Chronos(MAE)**|- |3/9|**7/9** |
> | **Mean rank(MAE,23models)** |11.6 |14.9 |**9.1** |
>
> We also compare two strong Time-MMD models, iTransformer and PatchTST, against each other under swapped contexts. We find that:
> * The average change in MAE with correct vs. swapped context is less than 1%.
> * ITransformer outperforms PatchTST *even with incorrect swapped contexts*.
>
> This suggests that **Time-MMD may not be suitable for evaluating whether models can leverage window-aligned contexts**.
>
> Running Direct Prompt on Time-MMD is prohibitively expensive, so broader cross-dataset validation beyond DoubleCast remains future work.
>
> ---
>
> ### Q1: Since the LLM has access to future time series during generation, how do you ensure it does not leak future information?
>
> The leakage is intentional. We explicitly generate oracle-style context containing future-relevant information, because the purpose of CAF is to test whether models can exploit such context at all. This mirrors scenario-based settings from related work [3]. We will **clarify this design choice in the revised paper and discuss the ecological validity trade-off**.
>
> [3] Williams, Andrew Robert, et al. "Context is Key: A Benchmark for Forecasting with Essential Textual Information." arXiv:2410.18959. ICML 2025.
>
> ### Q2: Does forecast improvement strictly demonstrate that the context is truly useful, rather than shortcut learning?
>
> Since the contexts are LLM-generated, they are more varied than template-based contexts, which hinders lexical shortcuts. However, the generated contexts are still not as varied as non-synthetic contexts, so we cannot rule out that DoubleCast exploits recurring phrase fragments to improve its forecasts. Measuring this effect is an **interesting direction for future work**.

---

> > ### Author Rebuttal · Reviewer_28cc · 2026-04-04
> >
> > I still find the level of novelty insufficient. The authors clarify that DoubleCast is not intended as the main contribution, and instead emphasize the explicit “generate-then-verify” framework. However, the generation appears largely similar to prior work Aurora, with limited differentiation. The main addition is the verification step using an LLM as a judge, which has become a common practice in recent LLM-based data synthesis pipelines. Therefore, I do not find this aspect sufficiently novel. Based on this, I maintain my original score.

---

> > > ### Author Response · Authors · 2026-04-07
> > >
> > > Thank you for your comments. Our paper differs from prior work in two key ways. First, we systematically evaluate whether DoubleCast truly leverages contextual information through both no-context and shuffled-context experiments. Second, our verification step goes beyond a simple LLM-as-a-Judge setup, where the model merely outputs a YES/NO decision on whether the context is complementary. Instead, we assess this quantitatively by measuring the change in CRPS for a GPT-5.2-based Direct Prompt forecaster. To our knowledge, this approach has not yet been used for a time series forecasting usecase.

---

### Official Review · Reviewer_8qSH · 2026-03-07

**Soundness:** 2
**Presentation:** 3
**Significance:** 2
**Originality:** 3
**Overall Recommendation:** 4
**Confidence:** 3

**Summary:**

This paper studies the problem of **context-aided forecasting (CAF)**, where natural language context is used to improve time-series prediction. The authors argue that existing multimodal datasets often suffer from weak or noisy context signals, making it difficult to reliably learn context utilization. To address this, they propose a semi-synthetic data generation methodology that uses large language models to create plausible textual contexts aligned with both historical and future time-series values, while verifying context usefulness through controlled forecasting comparisons.

Based on this pipeline, the paper introduces **CAF-7M**, a large-scale corpus containing 7 million context-augmented time-series windows across 11 domains, together with a rigorously curated test set. The authors further propose **DoubleCast**, a multimodal architecture designed to explicitly align textual and numerical modalities. A model trained solely on CAF-7M not only performs competitively on the synthetic benchmark but also generalizes to real-world datasets, demonstrating simulated-to-real transfer and effective context utilization.

**Compliance With Llm Reviewing Policy:**

Affirmed.

**Final Justification:**

Additional experiments and human annotations by rebuttal responded to my main questions, so I decided to improve my score.

**Key Questions For Authors:**

1. **Plans to address acknowledged limitations.**

    The paper discusses several limitations (e.g., unstable training data quality, limited diversity of synthetic text, weak temporal alignment, and limited long-horizon generalization). Could the authors elaborate on concrete plans to address these issues or provide additional analyses in the rebuttal?

2. **Cross-dataset validation.**

    Could the authors provide additional experiments comparing DoubleCast (or comparable models) trained on existing datasets to better support claims about dataset quality and generalization?

**Limitations:**

Yes

**Strengths And Weaknesses:**

Strengths

- **Clear problem formulation.** The paper identifies the challenge of weak or noisy textual context in multimodal time-series forecasting and formulates the context-aided forecasting setting clearly.
- **Principled data generation pipeline.** The semi-synthetic framework uses LLMs to generate context aligned with time-series signals and includes verification steps to ensure context usefulness.
- **Large-scale dataset contribution.** CAF-7M provides a sizable corpus covering multiple domains, which can support research on multimodal forecasting.
- **Model–data alignment.** The proposed DoubleCast architecture explicitly models interactions between textual and numerical modalities.


Weaknesses

- **Evaluation bias from LLM-based verifier.**

The test set is curated using a strong LLM verifier that retains samples where context improves prediction. This may bias evaluation toward context types favored by that specific model and affect fairness across model families.

- **Train–test distribution mismatch.**

The training data is not quality-filtered while the test set is strictly curated, creating a distribution gap where models learn from noisy contexts but are evaluated on high-quality ones.

- **Limited validation of synthetic context quality.**

Context quality is defined primarily by predictive improvement, without validating realism, causal plausibility, or alignment with real-world event distributions. Moreover, the claim that prior datasets suffer from weak or noisy context is not supported by systematic cross-dataset comparisons, as most quality analyses are limited to the proposed dataset.

- **Insufficient temporal grounding.**

The model lacks explicit mechanisms to align textual events with specific forecast horizons, which may dilute time-localized effects in time-series prediction.

- **Future-information leakage in context generation.**

Context is generated using both historical and future series values, which does not reflect real-world information availability and may affect ecological validity.

---

> ### Author Rebuttal · Authors · 2026-03-31
>
> Thank you for the detailed and constructive feedback. We address each point below.
>
> ---
> ### W1: Evaluation bias from LLM-based verifier.
>
> We agree this is a valid concern. Since filtering and context generation could favor GPT-5.2 and Llama-3.3, we ran the Figure 3 experiment to compare forecasting with either no context or the wrong context. On the Accepted set, **GPT-5.2 benefits slightly more** from the contexts, but Llama-3.3 and Qwen are close. On the Rejected set, all models incur **similar performance degradation**. This experiment cannot distinguish between verifier bias and a stronger GPT-5.2, which **we make explicit in the revised paper**.
>
>
> ### W2: Train–test distribution mismatch.
>
> We acknowledge this mismatch. This setup mirrors **common practice**: large models are trained on noisy data and evaluated on cleaner benchmarks. An in-domain validation experiment shows a clear **quantity-versus-quality trade-off**. In-domain validation cross-entropy at ~40k steps is shown below (lower is better):
>
> | Budget | filtered @ 40k | unfiltered @ 40k | gap (%) |
> | --- | ---: | ---: | ---: |
> | 50k | ~2.89 | ~2.95 | +2.0% |
> | 25k | ~3.10 | ~3.24 | +4.3% |
> | 10k | ~3.63 | ~3.75 | +3.2% |
> | 5k | ~4.11 | ~4.20 | +2.1% |
>
> Filtered data appears more sample-efficient, although **larger unfiltered sets can recover or exceed the gap** (e.g., `unfiltered_50k` outperforms `filtered_25k`). However, these runs used a cheaper LLM than the GPT-5.2-based filter, so we view this result as suggestive, not definitive. This trade-off is one reason we kept the training set unfiltered. We now **include these results in the appendix**.
>
> ### W3: Limited validation of synthetic context quality.
>
> Our primary quality criterion is *predictive utility*: context is useful if it measurably improves forecasts. We validate this mainly through the comparison with vs without context during verification, with the **shuffled-context** experiment providing an additional control showing that **gains depend on aligned context rather than generic text**. We also conducted a **human evaluation** with 11 annotators: majority-vote judgments were **positive for 80% of examples on background clarity and 82% on plausibility**. This human evaluation will be **included in the appendix**.
>
> ### W4: Claim about prior datasets having weak/noisy context is not systematically supported.
>
> We will **soften this claim** and ground it in prior literature. We find that **swapping context on Time-MMD instances changes MAE by less than 1%** on average for their top methods (iTransformer and PatchTST with BERT embeddings), which suggests that instance-specific context in Time-MMD has limited predictive value. This is consistent with prior work showing that benefits from context are not universal across datasets [1]. We acknowledge that **a broader, systematic cross-dataset comparison remains important future work**.
>
> [1] Zhang, Xiyuan, et al. "When Does Multimodality Lead to Better Time Series Forecasting?" arXiv:2506.21611.
>
> ### W5: Insufficient temporal grounding.
>
> This is a **limitation of DoubleCast rather than the CAF framework as a whole**. Direct Prompt performs well on CAF, suggesting that the generated contexts carry useful predictive signal. DoubleCast was intended primarily as a lightweight baseline to demonstrate CAF's value; **future work** should improve temporal alignment.
>
> ### W6: Future-information leakage in context generation.
>
> Thank you, this is intentional and we will state that more clearly in the revision. We use **offline future-conditioned generation to construct informative synthetic context** and test whether models can exploit free-form, forward-looking context at all. In that sense, previous work [2] similarly uses future-relevant textual context to evaluate context-aided forecasting. The main risk is therefore not accidental leakage, but **ecological validity** of the contexts, which we **acknowledge in the revised paper**.
>
> [2] Williams, Andrew et al. "Context is Key: A Benchmark for Forecasting with Essential Textual Information."  arXiv:2410.18959. ICML 2025.
>
> ---
>
> ### Q1: Plans to address acknowledged limitations.
>
> Our main next step is to **scale up the cross-verifier study** to better measure how much the accepted set depends on verifier choice or a larger, more diverse subset. Longer term, we will use this analysis to guide **multi-verifier filtering**, which can reduce single-verifier bias.
>
> ### Q2: Cross-dataset validation.
>
> We **evaluate DoubleCast on Time-MMD** both zero-shot and fine-tuned. Fine-tuning improves over zero-shot on 9/9 domains in MAE and 8/9 in MSE, and over Chronos on 7/9 domains in MAE and 8/9 in MSE. However, **Time-MMD appears to be more sensitive to the unimodal backbone than to multimodal capability**: iTransformer with swapped contexts outperforms PatchTST on 6 of 9 domains when both are equipped with BERT. Evaluating Direct Prompt on Time-MMD is prohibitively expensive.

---

> > ### Author Rebuttal · Reviewer_8qSH · 2026-04-01
> >
> > Thanks for the rebuttal so I will improve the score

---

> > > ### Author Response · Authors · 2026-04-07
> > >
> > > We sincerely thank you for your constructive engagement and for updating your score to reflect that your concerns have been fully resolved. We will be sure to incorporate these clarifications into the final manuscript.

---

### Decision · Program_Chairs · 2026-04-30

**Decision:**

Accept (regular)

**Comment:**

This paper proposes a context generation methodology for multimodal forecasting by prompting a Large Language Model (LLM) to generate plausible scenarios for time series datasets. This is used to create CAF-7M, a corpus of 7 million context-augmented time series windows. The authors train a fusion-based multimodal baseline on this corpus to validate that CAF-7M can serve as an effective context aided forecasting training dataset. This paper provides a principled  benchmarking framework  for both evaluating and training models for context-aided forecasting. This was a borderline-paper, and had mixed reviews, the major weaknesses mentioned by the reviewers included an evaluation-bias due to the GPT-based verifier,  not using multimodal TSFMs as verifiers,  train/test mismatch due to filtering only on the test set, and comparison against Aurora. The last weakness seems unfair given the recency of the Aurora paper and this paper’s greater focus on explicit verification The authors did address several of the other points via reasonable responses and more experiments/ablation. On the reviewer concerns around evaluation bias in test-set curation using GPT 5.2, this seems like a very consequential choice for the empirical conclusions of this paper, and deserves more rigorous evaluations to ensure there is no bias towards  context patterns favored by LLM-forecasters. While the authors did provide experiments with Direct Prompting using other LLMs, this question would be better addressed (as some reviewer suggested) via multiple different verifier architectures, and  a more comprehensive out-of-distribution benchmarking. Nevertheless, this work does make a very useful large-scale dataset and benchmarking contribution for research on multimodal forecasting.   I urge the authors to to address the weaknesses mentioned by the reviewers in a final version of the paper